# Boron Bioavailability Revisited: From Plasma-Accessible Species to Microbiota-Accessible Complexes—Implications for Nutritional Essentiality

**DOI:** 10.3390/biom15121711

**Published:** 2025-12-08

**Authors:** Andrei Biţă, Ion Romulus Scorei, Marvin A. Soriano-Ursúa, Cătălina Gabriela Pisoschi, Cristina Elena Biţă, Laura Dincă, Simona Ştefănescu, Maria-Victoria Racu, Iurie Pinzaru, Cristina Florescu, Diana-Ruxandra Hădăreanu, Cristian Adrian Siloşi, Johny Neamţu, Dan Ionuţ Gheonea, George Dan Mogoşanu, Marian Valentin Zorilă

**Affiliations:** 1Drug Research Center, Faculty of Pharmacy, University of Medicine and Pharmacy of Craiova, 2 Petru Rareş Street, 200349 Craiova, Romania; andrei.bita@umfcv.ro (A.B.); catalina.pisoschi@umfcv.ro (C.G.P.); johny.neamtu@umfcv.ro (J.N.); george.mogosanu@umfcv.ro (G.D.M.); 2Department of Pharmacognosy & Phytotherapy, Faculty of Pharmacy, University of Medicine and Pharmacy of Craiova, 2 Petru Rareş Street, 200349 Craiova, Romania; 3Department of Biochemistry, BioBoron Research Institute, S.C. Natural Research S.R.L., 31B Dunării Street, 207465 Podari, Romania; 4Department of Physiology, Escuela Superior de Medicina, Instituto Politécnico Nacional, Plan de San Luis y Diaz Mirón, 11340 Mexico City, Mexico; msoriano@ipn.mx; 5Department of Biochemistry, Faculty of Pharmacy, University of Medicine and Pharmacy of Craiova, 2 Petru Rareş Street, 200349 Craiova, Romania; 6Department of Rheumatology, Faculty of Medicine, University of Medicine and Pharmacy of Craiova, 2 Petru Rareş Street, 200349 Craiova, Romania; cristina.gofita@umfcv.ro; 7Department of International Relations, University of Medicine and Pharmacy of Craiova, 2 Petru Rareş Street, 200349 Craiova, Romania; laura.dinca@umfcv.ro; 8Clinical Laboratory, Emergency County Clinical Hospital of Craiova, 1 Tabaci Street, 200642 Craiova, Romania; simonastefanescubio@gmail.com; 9National Agency for Public Health, 67A Georghe Asachi Street, MD-2028 Chişinău, Moldova; maria-victoria.racu@ansp.gov.md (M.-V.R.); iurie.pinzaru@ansp.gov.md (I.P.); 10Department of Cardiology, Faculty of Medicine, University of Medicine and Pharmacy of Craiova, 2 Petru Rareş Street, 200349 Craiova, Romania; cristina.t.florescu@umfcv.ro (C.F.);; 11Department of Surgery, Faculty of Medicine, University of Medicine and Pharmacy of Craiova, 2 Petru Rareş Street, 200349 Craiova, Romania; cristian.silosi@umfcv.ro; 12Department of Physics, Faculty of Pharmacy, University of Medicine and Pharmacy of Craiova, 2 Petru Rareş Street, 200349 Craiova, Romania; 13Department of Gastroenterology, Research Center of Gastroenterology and Hepatology, University of Medicine and Pharmacy of Craiova, 2 Petru Rareş Street, 200349 Craiova, Romania; dan.gheonea@umfcv.ro; 14Department of Forensic Medicine, University of Medicine and Pharmacy of Craiova, 2 Petru Rareş Street, 200349 Craiova, Romania; valentin.zorila@umfcv.ro

**Keywords:** boron, microbiota-accessible boron complexes, plasma-accessible boron, quorum sensing, AI-2–borate, mucus barrier, essentiality, host–microbiota symbiosis

## Abstract

Boron (B) remains one of the least understood trace elements in human nutrition. Traditionally regarded as non-essential, its biological role has been reevaluated in light of emerging microbiome research. We provide a narrative synthesis of mechanistic, preclinical, and clinical studies to assess whether the colonic actions of B meet accepted criteria for nutritional essentiality. This review revisits B bioavailability through a dual-pathway framework distinguishing plasma-accessible boron (PAB)—small, fully absorbable species with transient systemic effects—from microbiota-accessible boron complexes (MABCs)—indigestible conjugates that reach the colon intact. Evidence indicates that PAB exerts short-term metabolic modulation, whereas MABCs act as prebiotic cofactors that stabilize microbial quorum sensing (autoinducer-2–borate; AI-2B), reinforce the colonic mucus barrier through borate–diol crosslinking, and support host–microbiota symbiosis. Deficiency or low intake of MABCs leads to dysbiosis, barrier fragility, and low-grade inflammation along gut–organ axes—effects reversible by MABC-rich diets. Analytical and clinical tools are proposed to discriminate between PAB and MABC pathways, including fecal B/speciation, AI-2B assays, and mucus-penetration markers. Recognizing B’s essentiality as a microbiota-dependent nutrient reframes its nutritional assessment, guiding future dietary guidelines and prebiotic design toward the microbiome–mucus interface.

## 1. Introduction

Boron (B) has long occupied an ambiguous position in human nutrition. Although B is widely present in plant foods and detectable in blood and tissues after ingestion, it has traditionally been considered non-essential for humans because no indispensable B-dependent host biomolecule has been conclusively identified and major authorities still do not recognize B as essential [1,2]. At the same time, observational and interventional findings associate dietary B with favorable outcomes across bone, immune, metabolic, and neurobehavioral domains, yielding a paradox: biologically active yet not essential under a host-centric definition. Here we revisit that paradox by reframing B bioavailability as two distinct access pathways with different ecologies and, we argue, different implications for essentiality [3,4].

The prevailing framework for essentiality emphasizes direct host-cell requirements and systemic bioavailability, privileging nutrients with intracellular targets and well-defined deficiency syndromes. Within this lens, fully absorbable, plasma-circulating B species—mainly boric acid (BA)—are rapidly taken up in the small intestine and cleared renally, showing largely transient systemic effects and lacking a canonical host-encoded cofactor [2,5]. We refer to these low-molecular-weight, diffusible species as plasma-accessible boron (PAB): forms that can modulate systemic processes (e.g., mineral metabolism, redox tone) but provide limited, short-lived availability to the colonic lumen.

Complementing PAB is a second, under-recognized route—microbiota-accessible boron complexes (MABCs)—comprising dietary or diet-derived B esters/chelates that resist digestion in the stomach and small intestine and reach the colon intact. In the colon, MABCs arise either from indigestible dietary conjugates (polyphenols, carbohydrates) or via in situ complexation of BA with organic ligands, where they interact with gut microbes, modulate community structure, and fulfill criteria of prebiotics [6,7,8]. Chemically, B is a Lewis acid that forms dynamic covalent bonds with *cis*-diols and catechols; as a result, it complexes with polyols, phenolics, and glycans to yield a family of organoboron conjugates (e.g., sugar–borate esters, B–pectic polysaccharides, B–phenolic acids) whose speciation depends on pH, ionic strength, and the local diol landscape. Unlike freely diffusible BA, these complexes are functionally gated by the colonic environment—microbial enzymes, mucin architecture, and the physicochemical milieu at the mucus–lumen interface. This site-specific accessibility reframes essentiality: not all ingested B reaches the same biological arena, and form determines function. In this review, MABCs denote B complexes that (i) are indigestible or poorly absorbed in the upper gut; (ii) are bioavailable to the colonic ecosystem (microbes and mucus); and (iii) can release, exchange, or shuttle B within the colon through reversible esterification/chelation [3,9,10,11].

This review evaluates whether B’s colonic actions, delivered via MABCs, fulfill accepted criteria for nutritional essentiality. We (i) define and contrast PAB vs. MABCs as access forms; (ii) synthesize mechanistic evidence at the microbiome–mucus interface; (iii) appraise animal and human findings through the lens of access form; and (iv) propose discriminative biomarkers (e.g., fecal B/speciation, autoinducer-2 (AI-2)/autoinducer-2–borate (AI-2B), mucus penetration) and study designs capable of testing essentiality. By adopting a symbiosis-aware framework, we aim to resolve past inconsistencies and clarify when and how B—specifically as MABCs—may qualify as a microbiota-dependent nutrient with actionable implications for dietary guidance and prebiotic design [2,3,4,6,8,12].

To facilitate reading, the paper progresses linearly from concept to application. Section 2 highlights the search methodology, inclusion/exclusion criteria, and Preferred Reporting Items for Systematic reviews and Meta-Analyses (PRISMA)-aligned structure. Section 3 formalizes the dual-access model distinguishing PAB from MABCs. Section 4 examines the molecular mechanisms within the colonic ecosystem, including quorum sensing (QS) chemistry, mucosal stabilization, and microbial metabolism. Section 5 reviews experimental and clinical evidence supporting these mechanisms. Building on this foundation, Section 6 and Section 7 address nutritional essentiality and translational implications—covering criteria, biomarkers, and perspectives relevant to dietary and regulatory science. Section 8 distills six key conclusions that collectively reframe B as a symbiosis-dependent nutrient.

## 2. Search Strategy and Selection Criteria (PRISMA-Inspired Approach)

This narrative review was informed by a structured literature search conducted in three major scientific databases (PubMed, Scopus, and Web of Science), covering the period January 1990 to February 2025. Although not a systematic review, the search and selection process adhered to core PRISMA principles, including a transparent search strategy, predefined eligibility criteria, screening process, and a structured approach to evidence synthesis [13].

Search strings were adapted to each database and combined Boolean operators with relevant biological and chemical terms, including: “boron”, “boric acid”, “borate esters”; “microbiota”, “gut microbiome”, “mucus barrier”, “mucin”; “AI-2”, “autoinducer-2”, “AI-2B”, “quorum sensing”; “short-chain fatty acids”, “SCFA”, “colon”, “colonic delivery”, “prebiotic”.

Studies were included if they met the following criteria: (i) biological relevance—data on B speciation, gastrointestinal (GI) fate, colonic accessibility, microbiota interactions, mucosal biology, QS, short-chain fatty acids (SCFAs), or host–microbiota symbiosis (HMS); (ii) model types—mechanistic in vitro studies, animal studies, or human studies; and (iii) outcome measures: effects on microbiota composition, mucus integrity, immune or metabolic markers, or systemic physiological endpoints.

Studies were excluded if they focused on non-B trace elements, lacked biological or mechanistic endpoints (e.g., purely analytical, geological, or industrial studies), addressed environmental or toxicological exposures unrelated to dietary B, and did not contribute to understanding host–microbiota interactions or colonic delivery.

Titles and abstracts identified through database searches were screened for relevance, followed by full-text assessment of eligible records. Additional articles were identified by reference mining of key publications and relevant reviews.

Included studies were categorized according to (i) B access form—PAB vs. MABCs, (ii) model system—in vitro, animal, or human, and (iii) primary biological outcomes—quorum signaling (AI-2/AI-2B), mucus barrier parameters, bacterial encroachment, SCFA profiles, microbiota composition, inflammatory or metabolic markers.

Evidence was synthesized narratively and integrated into the conceptual framework developed in Section 3, Section 4, Section 5 and Section 6, with explicit emphasis on distinguishing biological effects by B access form.

## 3. The Concept of Nutritional Essentiality

In nutritional science, essentiality denotes dietary substances that are indispensable for health, growth, development, or survival and cannot be synthesized endogenously in sufficient quantities [14]. This framework underpins dietary guidelines and nutrition policy worldwide and has historically been defined almost exclusively with respect to host cellular metabolism. A distinction is often drawn between nutritional essentiality and physiological essentiality [15]: nutritional essentiality implies dietary dependence, whereas physiological essentiality encompasses compounds required for biological processes regardless of endogenous synthesis. For example, cholesterol is physiologically essential as a membrane constituent and steroid precursor, yet not nutritionally essential because humans can synthesize adequate amounts de novo [16]. A compound is classified as nutritionally essential if sustained absence from the diet leads to identifiable dysfunction, pathology, or deficiency syndromes and if reintroduction restores normal biological function. On this basis, international regulatory bodies such as the *Codex Alimentarius* and the European Food Safety Authority (EFSA) have established nutrient reference values (NRVs) to define safe and adequate intake levels. Nutritional essentiality currently applies to four broad classes [14,17,18,19]:Water, as the universal solvent and medium for biochemical reactions;Macronutrients, including essential amino acids and essential fatty acids, required in gram quantities;Micronutrients, such as vitamins and minerals, needed in milligram or microgram quantities;Trace elements (e.g., Zn, Cu, I, Se, Mo, Cr, Si, Sn, V) that serve structural or catalytic roles despite low concentrations.

B has not been formally included in these categories, largely because no intracellular B-dependent enzymes or cofactors have been identified in humans. However, recent evidence challenges this host-centric paradigm by revealing critical roles for B at the microbiota–mucosal interface—raising the question of whether essentiality should also be evaluated within HMS [3,20].

Before revisiting classical definitions, it is necessary to clarify why essentiality must be considered in terms of access form—whether B reaches systemic circulation or the colonic ecosystem.

### 3.1. Why Essentiality Must Be Evaluated by Access Form

Classical definitions of nutritional essentiality are formally agnostic to anatomical site, but, in practice, they have been applied through a host-centric lens that emphasizes systemic bioavailability and intracellular targets as the primary arbiters of deficiency and adequacy [21]. Within this framework, nutrients are typically judged by their capacity to appear in plasma in a metabolically usable form and to correct overt deficiency syndromes.

Over the last decade, micronutrient bioavailability and gut microbiota (GM) science has shown that this view is incomplete. For many vitamins and minerals, the chemical form, food matrix, and site of release in the GI tract can profoundly alter absorption, tissue distribution, and biological effect [22,23,24,25]. In parallel, the concept of MACs has demonstrated that non-digestible substrates acting primarily in the colon can be critical for maintaining microbial ecology, barrier function, and host health, despite minimal direct systemic absorption [26,27,28,29].

By analogy, essentiality of B cannot be evaluated solely on the basis of small, freely diffusible species that enter the bloodstream. In this review, we therefore distinguish PAB—low-molecular-weight, fully absorbable species with predominantly transient systemic effects—from MABCs—indigestible or poorly absorbed conjugates that reach the colon and act locally at the microbiome–mucus interface. In this framework, form determines function: identical elemental intakes can yield different biological outcomes depending on chemical speciation and the anatomical site at which B becomes bioavailable, consistent with broader evidence that both chemical form and site of action shape the functional impact of nutrients and prebiotics [22,24,25,30].

Figure 1 provides a conceptual map of this dual-access model and orients to the operational distinctions detailed in Table 1 and the mechanistic development.

### 3.2. Criteria for Nutritional Essentiality

Widely accepted criteria for classifying a compound as nutritionally essential include [15,31,32,33,34]:It is indispensable for human survival, development, or health maintenance;Its absence from the diet produces a defined clinical or functional deficiency;It cannot be synthesized endogenously in sufficient amounts;Deficiency is reversible only through dietary intake of the compound (or a defined precursor);A dose–response relationship exists between intake and a measurable biological function.

Historically, essentiality has been established for water, macronutrients, micronutrients, and several trace elements; B has not been included due to the lack of a canonical host-encoded cofactor. These criteria apply irrespective of anatomical site—intracellular, systemic, or localized within the GI tract, including the GM and mucosal interface. This broader interpretation acknowledges that the HMS itself is a determinant of systemic health. Compounds not absorbed systemically but required for eubiosis, immune modulation, or barrier function, may therefore fulfill essentiality criteria. In this context, MABCs emerge as candidates for essential nutrients—not because they are required directly by host cells, but because they are indispensable for maintaining the microbial–mucosal ecosystem that supports host physiology.

### 3.3. Redefining Essentiality Through the Lens of Microbiota-Accessible Nutrients

The classical definition of nutritional essentiality has been host-centric, emphasizing nutrients required directly by human cells for survival, development, and physiological maintenance. Recent advances in microbiome science have catalyzed a paradigm shift: certain compounds—though not absorbed systemically—exert indispensable effects via their interaction with the GM. These compounds are increasingly recognized as functionally essential for preserving HMS, an axis fundamental to immunity, metabolism, and aging. Prebiotics exemplify this category and are defined as “selectively utilized substrates that confer a health benefit on the host by modulating the composition or activity of the gut microbiota” [35]. Such microbiota-accessible nutrients, resistant to enzymatic digestion in the upper GI tract, reach the colon intact where they support microbial diversity, promote SCFA production, and reinforce the colonic barrier.

Although most prebiotics (e.g., fibers, oligosaccharides, polyphenols) are not classified as essential in the traditional sense, their absence is increasingly linked to dysbiosis (DYS) and downstream pathologies (inflammatory bowel disease (IBD), metabolic syndrome, neuroinflammation, autoimmunity, accelerated aging) [7,36]. This evidence calls for evaluating essentiality not solely by systemic bioavailability, but by a compound’s capacity to sustain a healthy, functional microbiome. Under this expanded framework, a nutrient may be considered essential if it is required for health maintenance through microbiota-mediated mechanisms, is not synthesized in sufficient amounts by host or microbiota, leads to functional deficits when absent, cannot be replaced by systemically absorbed analogs, and exhibits a dose–response relationship linked to physiological outcomes.

This redefinition accommodates nutrients whose primary site of action is the colonic lumen, acting at the microbial–mucosal interface. In the case of B, the relevant forms are MABCs, whose reversible borate–diol chemistry enables local functions not reproduced by plasma B—see Figure 1 (dual-access model) and Table 1 (operational discriminators).

### 3.4. When Is Boron Essential? Clarifying the Role of MABCs

Traditionally, nutritional essentiality has been assessed through a nutrient’s direct effects on host cells and the presence of deficiency syndromes upon deprivation. However, this host-only perspective is challenged by evidence that GM is a functional partner in human physiology. For B, essentiality cannot be evaluated solely through systemic presence (e.g., plasma BA). Rather, it must be understood in terms of microbiota-mediated functions occurring at the colonic interface.

B becomes functionally essential when it reaches the colon in microbiota-accessible forms (MABCs), that is, indigestible dietary or diet-derived complexes capable of interacting with microbial and mucosal targets [3,6,8]. In contrast, PAB, though fully absorbed and systemically distributed, provides negligible delivery to the colon and thus does not elicit these localized effects.

Across multiple experimental models and human observations, benefits associated with eubiosis, mucus-barrier integrity, and metabolic balance are most consistently observed when B is delivered as MABCs acting in the colon, rather than as systemically absorbed PAB. Consequently, the health impact of B is conditional on microbiota-accessible delivery, and any rigorous assessment of essentiality should explicitly include site-specific accessibility as a defining criterion.

Operational discriminators between access forms are summarized in Table 1, which is referenced throughout this review when aligning mechanistic, preclinical, and clinical findings.

### 3.5. Mechanistic Preview—Why Form Determines Prebiotic Function (PAB vs. MABCs)

In the colonic environment, B’s Lewis acid chemistry underpins three interlocking mechanisms that are form dependent. First, B stabilizes AI-2B quorum signals, sustaining interspecies coordination in mixed consortia. Second, reversible borate–diol crosslinking with mucin O-glycans tightens the polymer network, reinforcing barrier thickness and limiting bacterial encroachment. Third, by supporting quorum capacity and barrier integrity within diol-rich matrices, MABCs bias metabolism toward a SCFA-centered ecology (acetate, propionate, butyrate (BUT)), with downstream effects on colonocyte energetics and host immunometabolism [37,38,39]. Complementing these local actions, evidence for reverse B trapping (bidirectional plasma–colon exchange) suggests that even plasma B can be recaptured in the lumen as borate esters; however, targeted colonic delivery via MABCs provides the most parsimonious route to achieve sustained, site-specific function. These mechanisms together explain why MABCs—rather than PAB—account for barrier integrity, microbial stability, and control of low-grade inflammation [8].

### 3.6. MABCs and Nutritional Essentiality—Conceptual Position and Testing Strategy

Nutritional essentiality requires that a dietary compound be indispensable for health, not produced endogenously in sufficient amounts, and that its absence leads to reversible dysfunction at physiological intakes. Within this framework, MABCs are best appraised as site-specific, form-dependent contributors to host physiology: their local functions in the colon—stabilization of AI-2B quorum signaling, reversible borate–diol crosslinking of mucins, and SCFA-centered metabolic ecology—are not reproduced by PAB under typical intakes. Thus, form determines function, and the relevant question is whether colonic access via MABCs is necessary to maintain eubiosis, barrier integrity, and low-grade inflammatory control that support systemic health [40,41,42,43,44,45,46].

Formal evaluation of MABC essentiality should rely on access form matched interventions (equal elemental B delivered as PAB vs. MABCs) and discriminative biomarkers aligned to mechanism: fecal B/speciation (colonic availability), AI-2/AI-2B capacity proxies, mucus penetration/thickness and bacterial encroachment, and SCFAs/compositional stability. Rescue specificity (benefit with MABCs but not with PAB at matched doses), dose–response for microbiota-accessible delivery, and reversibility of dysfunction with physiological intakes constitute the decisive tests. Collectively, these features justify evaluating B’s essentiality through microbiota-accessible delivery, with conclusions grounded in access form matched trials and the biomarker strategy outlined above.

## 4. Essentiality of MABCs for Normal Human Physiology

B is unequivocally essential in multiple non-human systems—vascular plants, algae, diatoms, and cyanobacteria—where it supports cell wall stabilization, reproductive development, and signaling [47,48]. In humans, essentiality remains formally unassigned because no B-containing enzyme or cofactor has been identified. Advances in microbiome science expand this assessment beyond intracellular targets to the HMS, where MABCs display their primary biological effects locally in the colon: microbial signaling, mucosal structure, and gut–organ homeostasis.

MABCs bypass digestion in the upper GI tract and reach the colon intact [6]. There, B’s Lewis acid chemistry enables reversible interactions with diol-rich matrices—mucins, microbial exopolysaccharides (EPS), and dietary polyphenols—supporting three interlinked functions: (i) stabilization of AI-2B QS signals that coordinate microbial communities; (ii) crosslinking of mucin glycans that strengthens the intestinal barrier; and (iii) promotion of a saccharolytic, SCFA-centered ecology that sustains colonic and immune balance [6]. MABCs may originate from dietary sources or form in situ via complexation of dietary B with indigestible ligands such as polyphenols and pectins [8]. The defining determinant is site-specific accessibility, not exogenous structure per se.

From this perspective: (i) B is not synthesized endogenously and cannot be substituted by another element in its biological roles; (ii) MABCs are not host-synthesized but can arise spontaneously from dietary B during GI transit; and (iii) B must reach the colon in a microbiota-accessible form (MAF) to fulfill its essential structural and signaling functions. Therefore, essentiality is conditional upon access form: identical elemental intakes can yield distinct outcomes depending on whether B is delivered as PAB or MABCs.

### 4.1. MABCs and Their Role in Growth, Health, and Survival

An expanding body of evidence indicates that MABCs act locally in the colon to stabilize microbial communication, reinforce the mucosal barrier, and sustain a resilient, metabolically balanced ecosystem across the lifespan. Mechanistically, MABCs (i) enable AI-2B-dependent quorum signaling in mixed consortia; (ii) crosslink mucin O-glycans through reversible borate–diol chemistry that tightens the mucus hydrogel; and (iii) bias community metabolism toward SCFA-centered saccharolysis that supports barrier maintenance and systemic immunometabolism [3,6,8,9,10].

In DYS, loss of these functions manifests as barrier thinning, bacterial encroachment, increased permeability, and low-grade inflammation—drivers implicated in metabolic, neuroinflammatory, and immune dysfunction [49,50]. In aging populations, where mucosal thinning and DYS are common, MABC-oriented strategies are particularly relevant to frailty and “inflammaging”, with signals linking MABC delivery to improvements in function and healthspan [8,44,51,52].

Biomarkers/predictions refer to: (i) increasing of fecal AI-2 and AI-2B bioassay activity; (ii) increasing of mucus thickness with decreasing of penetration and bacterial encroachment distance; (iii) increasing of SCFAs, especially BUT, with enrichment of butyrogenic guilds; and (iv) decreasing of intestinal permeability and inflammatory proxies. As an essentiality link, when these readouts are rescued by MABCs but not by matched PAB, the indispensable function is form- and site-specific.

### 4.2. Bidirectional Plasma–Colon Exchange in Boron Metabolism

In ruminant models, sustained oral B intake produces a progressive rise in fecal B that, after several days, can exceed urinary excretion despite rapid small-intestinal absorption and renal clearance [12]. This pattern supports reverse flux of neutral BA from plasma to the colonic lumen, followed by “B trapping” as indigestible borate esters with *cis*-diol-rich matrices (mucin O-glycans, dietary polyphenols, microbial EPS) [3,12].

Recognition of this bidirectional plasma–colon exchange challenges the conventional unidirectional model of B metabolism (diet → plasma → urine). It highlights that PAB is not metabolically inert with respect to the GM and that even fully absorbable B forms can be recycled into the colonic environment. However, from a physiological and safety perspective, this reverse transfer is not necessarily desirable. Prolonged presence of elevated B in the bloodstream increases tissue B loading, with evidence suggesting higher accumulation in women than in men, which may lead to organ-specific toxic effects over time [4,53]. Given that our proposed essential role of B lies in its MAFs within the colon, it is preferable to deliver B in indigestible, microbiota-accessible boron (MAB)-forming complexes directly to the large intestine. This targeted delivery minimizes systemic exposure, avoids unnecessary tissue accumulation, and ensures that B is available where it is nutritionally indispensable for maintaining HMS.

Considering biomarkers/predictions, (i) fecal B concentrations and complexed-species profiles increase with sustained intake; (ii) there is a positive coupling between fecal B, mucus integrity metrics and AI-2B capacity; and (iii) attenuation of fecal B rise when *cis*-diol-rich substrates are limited. Essentiality link emerges from the observation that reverse trapping provides a mechanistic basis for maintaining the colonic B required for quorum signaling and barrier functions; efficacy remains contingent on *cis*-diol-rich matrices—i.e., the MABC context.

### 4.3. Absence from the Diet or Inadequate Intake of MABCs

Insufficient delivery of MABCs impairs AI-2B-mediated coordination, weakens the mucin barrier, and increases epithelial permeability, thereby fueling DYS and low-grade systemic inflammation [3,6,49,50]. Downstream consequences extend across gut–organ axes—including musculoskeletal, neuroinflammatory, and metabolic domains—and are consistent with a deficiency phenotype centered on colonic function [50,54]. Ecological and clinical observations linking environmental/dietary B and joint outcomes (e.g., osteoarthritis (OA)) are hypothesis-generating for MABC dependence [55,56]. Figure 2 summarizes this functional cascade from reduced AI-2B capacity and barrier fragility to DYS, permeability, and inflammatory axes. In this context, prebiotic B deficiency may constitute a modifiable risk factor for microbial imbalance and chronic disease progression. Depletion of B within the colonic mucus layer has been associated with the overproduction of bacterial metabolites and proinflammatory mediators, including those implicated in OA pathophysiology. Supporting this mechanistic link, ecological and clinical studies have reported inverse correlations between environmental B levels and OA prevalence. Moreover, inadequate MABC intake may weaken the mucus barrier and disrupt beneficial microbial signaling, creating a feedback loop of microbial instability and host dysfunction [3,50].

Recent research further implicates DYS as a contributing factor in numerous extraintestinal disorders, particularly those involving the gut–brain and gut–joint axes. Conditions such as rheumatoid arthritis (RA), spondyloarthritis, and neuroinflammatory syndromes are increasingly associated with underlying microbial imbalances [57,58,59]. These observations converge on a central hypothesis: “Deficiency of MAB compromises microbial homeostasis and increases susceptibility to systemic inflammation, chronic disease, and age-related functional decline”. Within this framework, OA and RA may be considered clinical markers of chronic MAB insufficiency, particularly in older individuals. Additionally, MAB deficiency has been linked to decreased AI-2B-mediated signaling, reduced microbial diversity, increased intestinal permeability, and heightened proinflammatory tone.

A low-MABC state is expected to present with decreased AI-2/AI-2B signaling capacity, reduced microbial diversity and ecological stability, thinning of the mucus layer with deeper bacterial penetration, increased intestinal permeability, and elevated inflammatory biomarkers; composite symptom patterns along gut–joint and gut–brain axes may co-vary with these readouts (biomarkers/predictions).

A form-specific colonic deficiency that is reversed by MABC delivery—but not by matched PAB—supports symbiosis-dependent essentiality, i.e., B’s irreplaceable role is realized at the microbiome–mucus interface (essentiality link).

These mechanisms are integrated in Figure 3, which schematically illustrates how MAB operate at the microbiome–mucus interface. The figure connects three interdependent functions—AI-2B-mediated QS, mucin reinforcement through reversible borate–diol crosslinking, and SCFA-centered metabolic ecology—showing how local B availability orchestrates microbial stability, barrier integrity, and systemic homeostasis.

### 4.4. Boron Is Essential for Certain Physiological Functions

AI-2B forms when the bacterial precursor 4,5-dihydroxypentane-2,3-dione (DPD) complexes with borate; this interspecies quorum signal underpins community stability, biofilm architecture, and colonization resistance in mixed consortia [3,9,10]. In parallel, reversible borate–diol crosslinking tightens the mucin hydrogel, reducing bacterial encroachment and epithelial permeability and thereby stabilizing the barrier [3,6,8]. At both steps, B’s chemistry is non-redundant: (i) AI-2B requires borate at the complexation step, and (ii) the reversible borate–diol crosslinking is not substituted by other elements [60,61]. Because these functions are expressed locally at the microbiome–mucus interface, access form determines efficacy: MABCs provide sustained, on-site availability, whereas PAB is rapidly absorbed and cleared with limited lumen access.

Regarding biomarkers/predictions, the following observations can be highlighted: (i) fecal AI-2/AI-2B bioassay activity increases with local B availability; (ii) mucus thickness increases while penetration and encroachment indices decrease; and (iii) SCFAs—especially BUT—increase with enrichment of butyrogenic guilds as the ecosystem stabilizes.

The combination of non-redundant chemistry (AI-2B formation; borate–diol crosslinking) and rescue specificity by MABCs—but not by matched PAB—indicates a form- and site-dependent requirement to maintain normal colonic physiology and symbiosis (essentiality link).

### 4.5. Boron Deficiency Is Prevented by MABC Colonic Foods, Not by Other Elements

Because humans neither synthesize B nor retain it long-term, dietary delivery in MABC-forming structures (B–polyphenol esters, B–pectin complexes, sugar–borate esters) is required to support colonic functions [6,8]. Plasma-accessible forms (BA/soluble borates) are absorbed rapidly in the small intestine and typically do not reach the colon in functionally meaningful amounts. By contrast, indigestible complexes traverse to—and act within—the colonic lumen, engaging diol-rich matrices (mucins, microbial EPS, polyphenols) and sustaining quorum signaling and barrier stewardship. In practice, the boron nutrient density (BND) concept can help prioritize foods likely to deliver B to the colon rather than to plasma [6].

Biomarkers/predictions refer to (i) diets enriched in MABC-containing food matrices—fecal B and B-speciation indices higher; (ii) fecal AI-2/AI-2B bioassay capacity higher; (iii) mucus integrity higher (thickness) with lower penetration and encroachment and lower permeability proxies; (iv) inflammatory markers lower, in parallel with increases in SCFAs (particularly BUT) and enrichment of butyrogenic guilds.

Prevention or remission of the colonic deficiency phenotype (impaired AI-2B capacity, barrier fragility, DYS, increased permeability and inflammatory tone) by MABC-rich foods—but not by matched PAB—supports their status as functionally essential prebiotics (essentiality link).

The four mechanistic domains, their primary sites of action, discriminative biomarkers, and current evidence base are summarized in Table 2.

### 4.6. Critical Dietary Intake of MABCs

As with other essential nutrients, a functional threshold likely exists below which MABCs cannot sustain quorum capacity, barrier integrity, or SCFA homeostasis. Animal studies show intake–response relationships for fecal B, B associated with the colonic mucus gel, and histological measures of barrier integrity, consistent with form-dependent colonic bioactivity [6,8]. In human settings, OA has emerged as a condition sensitive to long-term B/MABC status, compatible with a dose–response at the symbiotic interface [56,62].

These observations indicate that adequacy—not merely presence—of MABCs is required to maintain HMS. Increased dietary delivery of MABCs associates with greater mucus resilience, lower intestinal permeability, reduced inflammatory tone, and restoration of microbial equilibrium [6,8].

The main biomarkers/predictions include (i) intake–response for fecal B and B speciation; (ii) concordant changes in AI-2/AI-2B bioassay capacity and mucus metrics (thickness, penetration, encroachment distance, permeability); and (iii) parallel movement of SCFAs (especially BUT) and inflammatory proxies.

Demonstrating a dose–response for access form matched MABCs satisfies a key criterion for essentiality, supporting their recognition as functionally indispensable prebiotic cofactors at the microbiome–mucus interface (essentiality link).

## 5. Physiological Response During Very Low Intake of Boron Complexes

The physiological actions of dietary B are governed by its biological access form, which determines whether the element acts systemically or symbiotically. Two principal routes are recognized: (i) PAB—small, digestible inorganic or low-molecular-weight organic species absorbed in the upper GI tract, acting systemically and transiently; and (ii) MABCs—indigestible conjugates that resist upper-gut digestion, reach the colon, and exert local functions at the microbiome–mucus interface.

Current evidence indicates that the MABC pathway satisfies nutritional essentiality criteria by sustaining HMS: reinforcing the mucus barrier (reversible borate–diol crosslinking), enabling interspecies quorum signaling via AI-2B, and modulating local immune–metabolic tone with downstream systemic effects. By contrast, the PAB route—while potentially beneficial in selected contexts (hormonal, enzymatic, redox, bone remodeling)—aligns with non-essential, pharmacological/modulatory actions under typical intakes [6,8].

### 5.1. Biological Effects of Low MAB Intake

A diet deficient in MABCs destabilizes the HMS and disrupts B-dependent QS (notably AI-2B), leading to a cascade of deficits across four interrelated domains:*Mucus barrier integrity*: the colonic mucus layer—essential for epithelial protection and immune surveillance—becomes structurally compromised in the absence of MABCs, weakening pathogen exclusion and metabolic homeostasis [63];*Mucosal defense*: reduced stimulation of BUT-producing bacteria, loss of B-mediated interbacterial signaling, and diminished borate–mucin crosslinking collectively impair colonization resistance [6,8];*Persistent DYS*: without adequate MABCs, the GM remains trapped in an imbalanced, aged-like state, accelerating immunosenescence and systemic functional decline [6,8];*Reduced resilience to stressors*: lower ecological stability of beneficial commensals increases susceptibility to antibiotics, pathogens, and oral–gut biofilm formation [6].

A low-MABC state is expected to present with decreased AI-2 and AI-2B signaling capacity, reduced microbial diversity and ecological stability, thinning of the mucus barrier with increased penetrability, and elevated intestinal permeability accompanied by higher inflammatory biomarkers (e.g., fecal calprotectin, high-sensitivity C-reactive protein (hs-CRP), interleukin-6 (IL-6)).

### 5.2. Sex-Specific Physiological Responses

Sex influences B kinetics, GM composition, and downstream physiological responses, potentially conditioning the efficacy of MABCs. Population studies report lower reference ranges for hair B in women compared with men, despite similar whole-blood levels—a pattern consistent with sex-specific tissue handling (distribution and excretion) rather than systemic exposure per se [53]. Such physiological differences may alter both peripheral and luminal B availability, particularly at the gut interface. Women also exhibit higher prevalence of DYS-associated conditions, including osteoporosis (OP), polycystic ovary syndrome (PCOS), ovarian cancer, and IBD—disorders increasingly linked to GM imbalance and disruption of B-dependent QS (AI-2B) and barrier homeostasis [3,4].

Across aging, sex remains a dominant covariate shaping GM structure and function, contributing to the so-called “male–female health–survival paradox”, in which women live longer yet experience greater frailty [64]. Men more frequently harbor *Lactobacillus gasseri*, *L. oris*, and *L. salivarius* together with proinflammatory taxa such as *Clostridium hathewayi* and *Eggerthella lenta*, whereas women show higher abundances of *Prevotella copri*, *Eubacterium rectale*, and *Roseburia inulinivorans*—species associated with barrier integrity and anti-inflammatory potential [65,66,67]. These ecological baselines plausibly modulate the magnitude of MABC effects, since MABCs act locally to reinforce the mucus barrier and stabilize interspecies signaling.

A further mechanistic layer involves hormone–microbiome crosstalk. Microbial β-glucuronidase activity governs enterohepatic estrogen recycling, while targeted modulation by *Lactobacillus* spp. and *Bifidobacterium* spp. can attenuate this process and help restore hormonal balance [68,69]. Dysregulated estrogen signaling is implicated across metabolic syndrome, breast and ovarian cancers, endometriosis, cardiovascular disease, and neurodegeneration. Notably, higher plasma B concentrations have been associated with increased all-cause mortality in women—but not in men—suggesting complex, sex-specific interactions between systemic B status and health outcomes [4].

Within this framework, fecal AI-2B capacity emerges as a promising non-invasive, sex-sensitive biomarker of luminal B availability and microbiota integrity [3]. Future interventional trials should pre-specify sex as a biological variable, integrate AI-2/AI-2B and barrier metrics, and test whether MABC-based interventions differentially promote healthspan in men and women (Table 3).

### 5.3. Dietary Delivery, Boron Nutrient Density, and the Symbiotic Role of MABCs

To operationalize colonic delivery, we distinguish total boron nutrient density (BND_T_)—total elemental B per 1000 kcal—from microbiota-accessible boron nutrient density (BND_MA_)—the indigestible fraction that reaches the colon and is usable by the GM [6]. In plant-based foods, ~10% of BND_T_ is typically microbiota-accessible, with higher proportions in polyphenol-rich matrices (notably phenolic acids) [6]. Accordingly, dietary patterns that prioritize indigestible B-binding structures—rather than total elemental intake alone—are most likely to deliver physiologically relevant B to the colon.

As a practical rule, achieving an effective colonic dose ≥ 1 mg MABCs/day generally requires BND_T_ ≈ 10 mg/day, attainable with ≥90% plant-forward diets rich in B–polyphenols and B–pectic polysaccharides (e.g., Mediterranean-style patterns) [8]. Consistently, epidemiological and experimental data show positive coupling between fiber-dense foods and total dietary B [44,70]. Major MABC donors include B–pectic polysaccharides [71,72], B–polyphenol esters/complexes [73,74], and emerging sugar–borate esters (e.g., lactose–borate) [75]. Diet analyses indicate the ~10% BND_MA_ fraction mirrors the average digestible B-carbohydrates/indigestible B–polyphenols ratio (~10:1), reinforcing the operational heuristic ≥ 10 mg/day BND_T_ → ≥1 mg/day MABCs [6,56]. As a feasibility/adherence marker, fecal B ≥ 1 mg/day has been proposed as a tractable biomarker of adequate MABC intake [6] (Table 4).

At the signaling level, MABCs supply B to stabilize AI-2B, sustaining cross-species quorum behaviors, microbial diversity, and colonization resistance [11]. At the structural level, borate–diol crosslinks with O-glycosylated mucins strengthen the mucus hydrogel and support immune–metabolic homeostasis [3,12]. This dual mechanism echoes B’s evolutionary roles in plant–microbe symbioses and, in humans, manifests as reduced barrier fragility and inflammatory tone, enhanced resilience to antibiotic perturbations, modulation of estrogen recycling via microbiome-dependent pathways, and strengthened mucosal immunity. Detection of B in the colonic mucus layer after supplementation further supports site-specific, symbiotic action [8].

Human and animal cells may not require B as a direct cytosolic cofactor; rather, the GM requires B to maintain a health-supporting symbiosis with the host. Consequently, the essential physiological impact of dietary B is realized through prebiotic action—stabilizing QS signaling and reinforcing the mucus barrier—provided delivery is microbiota-accessible. In practice, dietary design and prebiotic formulation should emphasize BND_MA_-rich foods and MABC donors, with fecal B/speciation and AI-2/AI-2B readouts as tractable markers of colonic delivery and effect (Table 4; Figure 3).

Increasing trajectories of total to microbiota-accessible BND (BND_T_ → BND_MA_) are expected to correlate with: (i) higher fecal B concentrations and complexed-species profiles; (ii) enhanced AI-2 and AI-2B signaling capacity; (iii) greater mucus thickness with reduced penetrability; (iv) decreased intestinal permeability and inflammatory biomarkers; and (v) elevated SCFA production (particularly BUT) accompanied by enrichment of butyrogenic microbial guilds.

## 6. Challenges and Future Directions for Recognizing Prebiotic MABCs as Nutritionally Essential

The colon is a strategic site for nutrient bioavailability: near-neutral pH, low host enzymatic activity, and longer transit collectively favor microbiota-accessible delivery [76]. Within this milieu, MABCs—indigestible B–diol conjugates—reach the large intestine intact and act locally at the microbiome–mucus interface (AI-2B quorum signaling, borate–diol mucin crosslinking, SCFA-centered ecology), with downstream effects along gut–organ axes [11]. These form- and site-dependent actions are not reproduced by fully absorbable, PAB under typical intakes. Practically, MABC-rich dietary patterns—pectins, polyphenols, plant proteins, fermented foods—mirror Mediterranean eating and support healthspan through microbial resilience, mucosal protection, and systemic homeostasis; high-B carriers include coffee, cocoa, raisins, berries, quinoa, legumes, nuts, and selected marine sources. Figure 4 provides a forward-looking synthesis of the strategic role of MABCs and Table 5 outlines measurable targets and candidate biomarkers for development programs.

### 6.1. Conceptual and Definitional Needs

A primary task is to operationalize essentiality by access form. Classical, host-centric frameworks emphasize systemically absorbed nutrients; MAB requires explicit recognition of site-specific essentiality in the colon. Consensus statements should (i) define a deficiency phenotype centered on reduced AI-2B capacity, barrier thinning/penetration, increased permeability, DYS, and low-grade inflammation; and (ii) specify rescue criteria that are unique to MAB (i.e., not reproduced by matched PAB) [6,8]. Cross-references to biomarker panels in Table 5 will standardize assessment.

### 6.2. Analytical and Speciation Needs

Discriminating MAB from PAB in complex matrices demands standardized, orthogonal analytics: fecal/colonic B quantification and speciation, validated AI-2/AI-2B bioassays, biophysical/imaging readouts of borate–diol crosslinking (mucus thickness, penetration indices, encroachment distance), and permeability proxies (e.g., lactulose/mannitol (L/M) ratio) [6]. Rigorous sample handling (pH, reversible ester dynamics) is essential to avoid artifactual loss of complexed B.

### 6.3. Clinical Trial Design and Causality

Causality hinges on access form matched interventions: equal elemental B delivered as PAB vs. MAB with colonic endpoints. Primary readouts should include fecal B/speciation, AI-2/AI-2B capacity, mucus thickness/penetration, permeability indices, SCFAs, and microbiome structure/stability. Factorial designs can test synergy with fibers/probiotics; dose–response should be referenced to the progression from BND_T_ to BND_MA_ for dietary feasibility. Trials should pre-specify sex as a biological variable and include follow-up sufficient to capture durable ecological change (Table 5).

### 6.4. Safety and Exposure Management

Because MABCs prioritize local action while limiting systemic exposure, safety appraisal should still consider sex differences in B handling, potential accumulation with chronic intake, and renal interactions [4,53]. Targeted colonic delivery is expected to mitigate tissue loading relative to fully absorbable PAB, yet nutrivigilance should track adverse events, endocrine endpoints, and mineral metabolism—especially in older or multimorbid cohorts.

### 6.5. Regulatory and Labeling Pathways

Current frameworks lack categories for microbiota-accessible essential nutrients. A pragmatic bridge is to position MABCs as prebiotics [35] with substantiation for barrier and microbiome-mediated outcomes, while building dossiers that distinguish MABCs from PAB in exposure and mode of action. Labeling could leverage BND_MA_ to communicate microbiota-accessible content and guide precision-nutrition planning.

### 6.6. Food Design and Dietary Patterns

Food design should favor matrices that protect B complexes through the upper GI and release them in the colon: pectin-rich fruits, polyphenol-dense plant foods, fermented products, and gentle processing that preserves diol-rich structures. Mediterranean-style patterns naturally deliver higher BND_MA_; targeted fortification with B–polyphenol/B–pectin complexes could close gaps in vulnerable groups. Candidate carriers include coffee/cocoa products, grape/berry preparations, legume flour, oat/β-glucan blends, and seaweed-based matrices.

### 6.7. Methodological Pitfalls and Confounders

Because MABCs co-occur with polyphenols and fibers that have independent bioactivity, confounding is a persistent risk. Trials should include PAB controls and matrix-matched placebos to isolate the B-specific effect. Given that AI-2B is ecology-sensitive, baseline microbiome stratification (e.g., enterotypes) and sex-specific analyses will improve interpretability [64,67].

### 6.8. Outlook

The future of B nutrition lies in prebiotic, microbiota-accessible delivery. Consistent MABC exposure appears necessary for eubiosis, barrier stewardship, and systemic homeostasis, whereas inadequate intake aligns with DYS and inflammatory trajectories. Resolving the analytical, clinical, and regulatory tasks above will enable formal recognition of MABCs as nutritionally essential and accelerate microbiota-targeted dietary and nutraceutical solutions across the lifespan.

## 7. Clinical and Regulatory Implications

Recognition of MABCs as nutritionally relevant introduces opportunities for clinical practice and public health, while also challenging host-centric nutrient frameworks. Most risk assessments and dietary reference values were derived for PAB (e.g., BA, inorganic borates), without considering the distinct metabolic fate, site of action, and exposure profile of MABCs. Below, we outline a translation pathway that is (i) access form matched, (ii) biomarker-anchored, and (iii) compatible with existing prebiotic standards [35], while distinguishing MABCs from fully absorbable B forms.

### 7.1. Clinical Nutrition: Candidate Indications, Phenotypes, and Readouts

The clinical rationale for MABCs stems from their ability to act locally at the microbiome–mucus interface, a critical site where molecular and ecological integrity determine systemic outcomes. Through reversible borate–diol crosslinking, MABCs strengthen the colonic mucus barrier and limit bacterial encroachment, while their participation in AI-2B quorum signaling supports interspecies microbial coordination. Together, these actions stabilize mucosal architecture, favor SCFA-centered metabolism, and modulate communication across multiple gut–organ axes, including gut–brain, gut–bone, and gut–joint pathways [3,6,8,11]. Clinically, these mechanisms converge on phenotypes defined by barrier fragility, DYS, and low-grade inflammation—conditions that often represent early or subclinical states of chronic disease.

Several candidate indications emerge from this mechanistic framework. First, barrier-fragility syndromes such as irritable bowel syndrome (IBS), with increased intestinal permeability, quiescent or mild IBD during maintenance phases, and DYS-associated constipation could benefit from barrier reinforcement and microbiota stabilization [3,63]. Second, inflammaging and frailty represent another promising domain, particularly in older adults with chronic low-grade inflammation, sarcopenia risk, or multimorbidity, where colonic mucus thinning and microbial imbalance are prevalent [8,44,52]. Third, the gut–joint and gut–brain axes provide emerging clinical frontiers, as both OA and neuroinflammatory syndromes demonstrate tight correlations between DYS, mucosal permeability, and symptom progression [50,55,56,57,58,59]. Finally, mucosal surfaces beyond the colon—including oral, vaginal, and dermal microbiomes—may share analogous diol-rich environments where B–polyol chemistry operates, suggesting a translational research avenue for extraintestinal applications of MABCs.

To evaluate efficacy, clinical readouts must correspond to the access form of delivered B. Primary indicators of colonic exposure include fecal B concentration and speciation profiles, with the presence of complexed species confirming microbiota-accessible delivery. Quorum signaling activity can be monitored through fecal AI-2 and AI-2B assays, where increased signaling capacity denotes microbial eubiosis restoration. Barrier integrity may be assessed by histological or imaging measurements of mucus thickness and penetration, permeability indices such as the L/M ratio, zonulin proxies, and bacterial encroachment distance. Finally, ecosystem and host tone can be captured through fecal SCFA profiling—especially BUT levels—quantification of butyrogenic taxa (e.g., *Faecalibacterium*, *Roseburia*), and inflammatory markers such as fecal calprotectin, serum hs-CRP, and IL-6.

In clinical practice, the decisive hallmark of MABC-specific activity lies in rescue specificity: when equal elemental B doses delivered as MABCs vs. PAB produce divergent outcomes on colonic endpoints such as AI-2B capacity, mucus integrity, and SCFA generation. This divergence constitutes the functional signature of form- and site-dependent essentiality and provides the clearest experimental criterion distinguishing nutritional from purely pharmacological effects of B.

### 7.2. Safety and Toxicology: Exposure Management for Access Form Specific Boron

Safety assessment of B must be anchored in the biology of access form. PAB and MABCs differ in chemistry, site of action, and exposure kinetics: PAB species (e.g., BA, borate salts, small B–carbohydrate adducts) are rapidly absorbed in the upper GI tract, produce transient systemic peaks, and are cleared primarily via the kidney; by contrast, MABCs are chemically complex, largely indigestible conjugates that act predominantly within the colonic lumen and, at dietary intakes, exhibit minimal persistence in blood or tissues [3,6,8]. These distinctions argue strongly against unitary toxicological treatment for all B species.

An access form aware risk-management framework follows. First, exposure classes should be differentiated: classical toxicology thresholds and systemic tolerable upper levels remain appropriate for fully absorbable PAB, whereas MABCs should be evaluated under a colon-localized exposure model that emphasizes fecal B quantification and speciation profiles over plasma accumulation. This approach reflects the intended site of action and the low systemic bioavailability of MABCs at dietary doses [4,53]. Second, sex must be treated as a biological variable in both safety monitoring and interpretation. Population data showing sex-specific differences in tissue handling of B (e.g., lower hair B reference ranges in women despite similar whole-blood values) indicate that distribution and excretion may differ by sex even when systemic exposure is comparable; prospective analyses should therefore predefine sex-stratified safety endpoints and subgroup analyses [4,53,64,65,66,67]. Third, nutrivigilance plans should include renal function monitoring where clinically appropriate, surveillance of endocrine and mineral metabolism during longer interventions, and the use of a standardized adverse-event lexicon tailored to prebiotic trials, recognizing that lumen-acting agents may shift stool characteristics or GI comfort without implying systemic toxicity. Finally, assay discipline is essential: because borate esters are reversible and pH-sensitive, specimen collection and handling must control pH and ionic conditions to avoid artifactual de-complexation and misestimation of the complexed fraction in speciation assays [6].

Current regulatory thresholds for B intake, including tolerable upper intake levels (ULs), are largely extrapolated from studies employing inorganic BA or borax administered systemically. Such data are informative for PAB but are not directly transposable to MABCs, which differ in pharmacokinetics, tissue distribution, and toxicity profiles. At dietary intakes, MABCs are designed to act locally in the colon, lack evidence of blood or tissue accumulation, and therefore warrant safety evaluation anchored to colonic exposure rather than systemic burden. Conflating all B forms under a single systemic UL is thus scientifically imprecise and risks imposing unnecessary restrictions on MABC-rich foods and formulations, while failing to maintain appropriate safeguards for high-exposure PAB scenarios. A future-proof regulatory framework should explicitly distinguish digestible, systemically bioavailable PAB species (subject to classical systemic toxicology constraints) from indigestible, prebiotic MABCs (evaluated with colon-centric exposure metrics and readouts). Such differentiation would better reflect real-world exposure, align safety endpoints with mechanism and site of action, and support proportionate, evidence-based guidance for both categories at dietary doses.

### 7.3. Dietary Guidance and Labeling: From Elemental Intake to Colonic Availability

Dietary guidance for B should pivot from total elemental intake to colonic availability, because symbiosis-dependent functions are governed by the fraction that reaches and acts at the microbiome–mucus interface. This can be operationalized with two complementary density metrics expressed per 1000 kcal: BND_T_, which captures total dietary B, and BND_MA_, which reflects the indigestible, lumen-available fraction usable by the GM [6]. In plant-forward dietary patterns, BND_T_ values of approximately 10 mg/day generally correspond to BND_MA_ deliveries at or above ~1 mg/day, a level consistent with colonic exposure targets [6,8,44,70].

In practice, exposure and effect should be tracked with coherent, access form specific readouts. As an adherence marker, fecal B ≥ 1 mg/day accompanied by a complexed speciation signature indicates adequate colonic delivery and retention of MAFs [6]. As mechanistic effect markers, increases in AI-2/AI-2B capacity, thicker and less penetrable mucus (with reduced encroachment), lower permeability and inflammatory proxies, and higher SCFAs—particularly BUT, together with enrichment of butyrogenic guilds—provide a rational panel for monitoring biological response. These endpoints align the dietary construct (BND_T_ → BND_MA_) with the intended site of action and the expected physiology of barrier stewardship and microbial eubiosis.

Labeling and communication should mirror this access form logic. Beyond listing total B, products intended to deliver lumen-acting species ought to declare “MABC, mg per serving” alongside BND_T_, with values substantiated by validated speciation and stability testing. A research-ready colonic delivery factor (CDF), a standardized index derived from in vitro stability and in vivo speciation data, could quantify the proportion of labeled B expected to reach the colon as complexed, MAFs. Structure/function statements should be framed around barrier integrity and microbiome-mediated endpoints (e.g., support for mucus layer integrity, quorum signaling capacity, and SCFA ecology), consistent with established definitions of prebiotics and within prevailing regulatory guidance [35].

Taken together, a shift to BND_MA_-aware guidance, paired with transparent labeling and access form specific biomarkers, provides a tractable pathway from formulation to physiological relevance, enabling both clinical translation and consumer-facing clarity.

### 7.4. Regulatory Pathways: Positioning MABCs Within Existing Schemes

The regulatory integration of MABCs requires a conceptual and procedural shift from systemic toxicology toward colon-localized, access form specific evaluation. In the near term, MABCs can be appropriately classified within the framework of prebiotic compounds, emphasizing their localized mechanism of action at the microbiome–mucus interface and their dependence on microbiota-mediated processes for efficacy [35]. This positioning avoids conflation with PAB and precludes pharmacological or disease-treatment claims, anchoring instead on nutritional modulation, barrier support, and microbiome symbiosis.

A scientifically robust regulatory dossier for MABCs must reflect this access form awareness across all evidence tiers.

First, under chemistry and manufacturing, regulatory documentation should include detailed conjugate characterization, with explicit management of reversible borate–diol ester dynamics and stability testing under GI pH conditions. Batch-level speciation fingerprints, showing the proportion of complexed vs. free B, should be part of standard quality control.

Second, exposure modeling must depart from the conventional plasma-based paradigm. Instead of systemic B levels, the principal exposure metric should be fecal B content and speciation, quantified relative to dietary intake and the BND_T_ → BND_MA_ progression. This approach reflects the colonic site of action and aligns risk–benefit evaluation with actual biological exposure.

Third, mechanistic substantiation should rely on endpoints consistent with MABC biology: (i) AI-2 and AI-2B QS assays as molecular readouts of microbial communication; (ii) mucus crosslinking indices and histological permeability measures as barrier integrity markers; and (iii) fecal SCFA profiling as an ecological biomarker of microbiota function. These assays, combined with predefined primary outcomes, form the mechanistic backbone for substantiation of prebiotic activity.

Fourth, safety evaluation should bridge toxicology to colon-localized exposure. Given the minimal systemic bioavailability of MABCs, safety dossiers should integrate sex-stratified analyses of renal and endocrine parameters, impurity and contaminant control, and long-term nutrivigilance plans addressing colonic tolerance rather than systemic accumulation [4,35].

Finally, clinical substantiation should rely on access form matched randomized controlled trials directly comparing MABCs with PAB at equivalent elemental B doses. Trials should prioritize colonic endpoints—such as AI-2B signaling capacity, mucus barrier thickness, and fecal B speciation—and include durability follow-up to assess sustained ecological and physiological effects.

This differentiated regulatory path would prevent inappropriate restrictions on MABC-rich foods or supplements based on systemic PAB toxicology, while ensuring that claims remain aligned with established prebiotic paradigms. By explicitly linking chemistry, exposure, mechanism, and outcome to site of action, regulators can more accurately capture the nutritional—not pharmacological—nature of MABCs.

The paradigm shift recognizing B’s prebiotic, symbiosis-dependent essentiality opens wide avenues for innovation. Functional foods, such as B-enriched fermented vegetables, fruit beverages, and polyphenol-rich plant extracts, can be reformulated to enhance MABC delivery. In medical nutrition, tailored MABC supplementation may offer benefit for older adults or patients with inflammatory bowel conditions, while nutraceutical development can explore B–polyphenol complexes designed for targeted colonic release. To support this transition from discovery to application, regulatory harmonization, professional education, and transparent benefit–risk assessments will be essential. These coordinated efforts will enable the safe and evidence-based translation of MABC science into future public health and clinical practice frameworks, ensuring B’s rightful place among the emerging class of microbiota-dependent essential nutrients.

### 7.5. Food Design, Medical Nutrition, and Product Development

Translating the scientific concept of MABCs into practical nutrition represents a genuine paradigm shift—one that redefines the objectives of food design, processing, and supplementation. Traditional fortification programs are guided by the goal of increasing plasma-accessible micronutrients (PAB), optimized for rapid absorption in the upper GI tract. In contrast, MABC-centered nutrition focuses on colonic bioaccessibility—the controlled delivery of B in stable, indigestible complexes that reach the microbiome–mucus interface intact and biologically functional. This approach requires rethinking both technological processes and regulatory standards to ensure that B’s prebiotic potential is fully realized.

#### 7.5.1. Design Principles and Technological Implications: Protection Through the Upper Gastrointestinal Tract

For MABCs to act effectively in the colon, they must survive gastric acidity and enzymatic hydrolysis in the small intestine. Structural embedding of B within diol-rich plant matrices—such as pectins, arabinogalactans, lignin–polyphenol conjugates, and glycosylated fragments of cell wall polysaccharides—provides this protection. These matrices act as molecular shields, maintaining the stability of borate–diol linkages until the complexes reach the distal gut. There, microbial fermentation, enzymatic deglycosylation, and near-neutral pH conditions favor the selective release and activation of B at its intended site of action [71,72,73,74,75].

#### 7.5.2. Preservation of Functional Diol Scaffolds During Processing

Maintaining the integrity of *cis*-diol-containing structures is essential for preserving B’s capacity to form reversible covalent bonds. Therefore, gentle processing technologies—neutral pH, minimal heat, and low oxidative stress—should replace conventional high-temperature or alkaline treatments that destroy functional diol moieties and yield nonfunctional, absorbable PAB species. Techniques such as low-temperature spray drying, vacuum dehydration, encapsulation in resistant starches, or cryogenic drying can stabilize borate–diol conjugates, ensuring retention of their MAF in both functional foods and nutraceuticals.

#### 7.5.3. Speciation-Guided Quality Control

A cornerstone of credible MABC production is analytical validation through B speciation assays, using ultra-high-performance liquid chromatography–mass spectrometry (UHPLC–MS), nuclear magnetic resonance (NMR), or isotopic tracing to confirm the presence and proportion of complexed vs. free B. Routine speciation profiling should be adopted as a regulatory standard, guaranteeing batch-to-batch consistency and functional authenticity. This would support the establishment of a CDF—a quantitative index expressing the proportion of total B that remains complexed and reaches the colon intact, analogous to established bioavailability coefficients in vitamin and mineral regulation.

#### 7.5.4. Mechanistic Co-Formulation Strategies

The functionality of MABCs can be enhanced by co-formulation with prebiotic fibers and probiotic strains that share mechanistic synergies. Fibers such as inulin or β-glucans can reinforce B retention and modulate microbial fermentation patterns, while selected probiotics (e.g., *Lactobacillus plantarum*, *Bifidobacterium longum*) may amplify AI-2B-mediated quorum signaling and stabilize mucosal ecosystems. However, these designs must be guided by factorial experimental frameworks that differentiate B-specific effects from broader microbiome modulators, maintaining scientific clarity and avoiding over-complexed formulations.

#### 7.5.5. Candidate Dietary Carriers

Multiple empirical data sources identify food matrices that are naturally enriched in MABCs or their biochemical precursors:Polyphenol-rich sources such as coffee, cocoa, grape skins, berries, and raisins are known to form stable B–polyphenol esters (e.g., diester chlorogenoborates), offering both antioxidant potential and B delivery capacity;Pectin-dense foods including citrus fruits, apples, and legumes contain B–pectate linkages, providing slow-release reservoirs that align with colonic fermentation kinetics;Cereal and legume derivatives, notably oat β-glucans and chickpea or soybean flours, exhibit favorable BND_T_/BND_MA_ ratios, balancing total B content with microbial accessibility;Marine polysaccharides such as laminarins and ulvans can stabilize borate esters under physiological pH, representing promising scaffolds for next-generation medical nutrition and functional formulations.

These carriers combine chemical resilience with biological relevance, offering both stability during processing and efficacy at the site of action. Their incorporation into dietary strategies, medical nutrition therapies, or nutraceutical formulations could support mucus-barrier repair, microbial homeostasis, and systemic anti-inflammatory balance in aging or vulnerable populations.

The food design of the future must move beyond fortification toward form-specific functionality, ensuring that B, and other prebiotic trace elements, are delivered in the molecular formats required to sustain HMS. This access form based design logic represents a decisive evolution in nutritional science, bridging molecular chemistry, microbiology, and clinical nutrition into a coherent, healthspan-oriented framework.

### 7.6. Trial Design and Causality: A Clinically Actionable Template

Establishing the nutritional essentiality of MABCs in humans demands a new generation of rigorously controlled form-specific comparative trials. These studies must explicitly differentiate between PAB and MAB exposure, recognizing that these two chemical pathways diverge fundamentally in absorption, distribution, and biological outcome. The decisive test of essentiality is rescuing specificity—the demonstration that physiological deficits associated with low B intake can be corrected by MABC supplementation but not by equivalent elemental doses of PAB.

#### 7.6.1. Experimental Framework

The core experimental architecture should deploy matched elemental B doses delivered in two standardized formulations: one fully digestible and systemically bioavailable (PAB), the other indigestible and colonic-accessible (MABC). This structure isolates the biological consequences of access form, allowing causal inference about B’s site-dependent functionality.

Primary endpoints must emphasize colonic and microbial outcomes, directly aligned with MABC mechanisms of action:Fecal B concentration and speciation profiles, distinguishing complexed vs. uncomplexed fractions to verify true colonic delivery;AI-2/AI-2B QS capacity, quantified through orthogonal bioassays, as a functional marker of microbial communication integrity;Mucus thickness, bacterial encroachment distance, and epithelial permeability, assessed via histological imaging and physiological markers such as the L/M ratio or zonulin levels;SCFAs, especially BUT, measured by gas chromatography (GC) or NMR as indices of microbial metabolism and barrier energy homeostasis;Microbial community structure, characterized by α- and β-diversity metrics and the abundance of butyrogenic taxa (*Faecalibacterium prausnitzii*, *Roseburia* spp., *Eubacte-rium rectale*);Inflammatory proxies, including fecal calprotectin, serum hs-CRP, and IL-6, which reflect systemic translation of gut barrier and immune modulation.

Secondary endpoints should extend to clinically relevant parameters such as IBS-type symptom composites, bone turnover markers, or systemic immune–metabolic indices, depending on the target population and the intended application (e.g., aging, inflammation, frailty).

#### 7.6.2. Design Features Ensuring Rigor

Scientific credibility and reproducibility require meticulous attention to study design parameters that address both biological variability and ecological confounding: (i) sex-stratified randomization and analysis should be mandatory, acknowledging well-documented sex differences in B absorption, metabolism, and physiological outcomes; (ii) baseline microbiome stratification by enterotype or diversity quartiles should be implemented to mitigate ecological bias and enable within-stratum causal inference [64,67]; (iii) matrix-matched placebos and PAB controls are essential to isolate B’s form-dependent effects from those arising due to co-nutrients, matrix components, or fermentation substrates; (iv) follow-up duration must be sufficient to capture durable microbial and mucosal remodeling; interventions of at least 12 weeks are recommended to ensure stabilization of community-level effects; and (v) pre-registration of analytical pipelines for AI-2B assays, mucus imaging, and SCFA quantification should be required to safeguard transparency and methodological consistency across research centers.

#### 7.6.3. Integration and Alignment

The parameters outlined above are anchored to the mechanistic and analytical framework established earlier in this review: intake targets, biomarker assays, mechanistic discriminators and evidence tiers. Collectively, these elements provide a translational roadmap from molecular insight to clinical validation. Well-designed, access form matched trials have the potential to decisively establish causality—demonstrating that MABCs, rather than systemically absorbed PAB forms, are the active agents in maintaining microbial homeostasis, mucosal integrity, and systemic anti-inflammatory balance. Such evidence would mark a critical inflection point in B nutrition research, advancing MABCs from theoretical constructs to formally recognized symbiosis-dependent essential nutrients.

### 7.7. Methodological Pitfalls and Confounders

Advancing the scientific validation of MABCs necessitates methodological precision across chemistry, biology, and ecology. Because MABCs occupy a unique position at the interface of inorganic chemistry and microbial physiology, their study is particularly vulnerable to experimental artifacts and ecological variability. Failure to address these issues can obscure true causal effects or generate misleading interpretations of B’s bioactivity. The following methodological domains warrant explicit standardization in future research.

#### 7.7.1. Speciation Artifacts and Analytical Stability

The reversible covalency of borate–diol esters underlies the very functionality that defines MABCs, yet this same property makes B speciation exceptionally sensitive to experimental conditions. Both sample collection and analysis must be carried out under tightly controlled parameters, typically at near-neutral pH (~7.0), physiological ionic strength, and minimal oxidative stress, to prevent artificial de-complexation, condensation, or ligand exchange. Small shifts in pH or redox potential can drive equilibrium toward free BA or borate anion (B(OH)_4_^–^), producing systematic underestimation of the complexed fraction.

Conversely, prolonged storage, freeze–thaw cycles, or exposure to metal ions can catalyze secondary condensation reactions, yielding spurious polymeric species. These distort both speciation profiles and quantitative B recovery. Analytical workflows should therefore employ stabilized buffers, chelator-free plastics, and prompt processing (< 30 min post-collection). Orthogonal methods, such as UHPLC–MS, NMR, and isotopic tracing, should be used in parallel to confirm the integrity of the MABC fraction and to quantify reversible equilibria [6]. Without such controls, interstudy comparisons remain unreliable and the physiological significance of MABCs cannot be meaningfully established.

#### 7.7.2. Co-Nutrient Interference and Matrix Complexity

Because MABCs arise naturally within polyphenol- and fiber-rich matrices, disentangling B-specific actions from co-nutrient effects poses a persistent challenge. Polyphenols, flavonoids, and prebiotic fibers possess intrinsic biological activities—antioxidant, fermentative, or signaling—that may independently modulate microbiome composition or mucosal physiology. These overlapping actions risk confounding the attribution of observed effects solely to B complexes.

To overcome this, matrix-matched controls and factorial study designs should be employed. For instance, comparing polyphenol matrices with and without B enrichment, or fibers with differential B-binding capacities, can help isolate form-dependent contributions. Furthermore, orthogonal statistical modeling (e.g., partial least squares regression or multilevel mixed-effects models) should be integrated to separate variance attributable to B speciation from that arising from co-metabolites. Only through such multifactorial approaches can the distinctive mechanistic role of MABCs—as quorum cofactors and mucosal stabilizers—be distinguished from the general bioactivity of their carrier matrices.

#### 7.7.3. Ecological Sensitivity of AI-2B Signaling

AI-2B signaling system provides a functional readout of microbial cooperation and eubiosis, yet it is inherently dynamic and ecologically sensitive. AI-2B levels fluctuate with microbial composition, substrate availability, antibiotic exposure, and even circadian rhythms. Inter-individual variability in diet or microbiome baseline can mask intervention effects if not properly accounted for.

To improve interpretability, clinical and nutritional studies must adopt standardized ecological baselines:Predefine antibiotic exclusion windows (typically ≥8 weeks) and document any anti-microbial exposures;Maintain controlled dietary records throughout the intervention period, with explicit tracking of fiber, polyphenol, and B intake;Perform baseline microbial sequencing (16S ribosomal ribonucleic acid (rRNA) or shotgun metagenomics) to classify participants by enterotype or diversity quartile, ensuring that AI-2B responses are contextualized relative to the underlying community structure.

Furthermore, parallel measurement of fecal SCFAs, mucus penetration indices, and inflammation markers provide a multidimensional framework for interpreting AI-2B dynamics. Without this ecological triangulation, AI-2B readouts risk being overinterpreted as causal when they may merely reflect transient compositional shifts.

#### 7.7.4. Translational Extrapolation and Cross-Species Scaling

Translating insights from preclinical models to human physiology requires caution. Rodent studies often overestimate systemic B exposure due to shorter intestinal length, faster transit, and simplified microbial consortia, which alter MABC persistence and function. Moreover, species differ in B turnover rates, mucin composition, and AI-2B regulatory networks, affecting the reproducibility of results across taxa.

Therefore, translational models must incorporate both elemental kinetics and chemical stability of B complexes. Comparative studies should use standardized in vitro gut simulations (e.g., simulator of the human intestinal microbial ecosystem (SHIME), TNO in vitro model of the colon (TIM-2)) to benchmark MABC degradation and release kinetics under human-relevant conditions. In vivo, physiologically-based pharmacokinetic models integrating colon pH gradients, microbial density distributions, and borate–diol dissociation constants can refine dose extrapolations between species.

Ultimately, the most reliable translation arises from human pilot trials employing ecological and chemical endpoints concurrently, bridging laboratory precision with biological complexity.

The rigorous study of MABCs demands a multidisciplinary methodological discipline that integrates chemical stabilization, ecological control, and translational modeling. By addressing these pitfalls proactively, researchers can move from associative observations toward mechanistically grounded evidence, establishing B’s role not merely as a trace element but as a form-specific, symbiosis-dependent nutrient essential for mucosal and systemic health.

### 7.8. Outlook: From Nutrient to Symbiotic Catalyst

The accumulating body of evidence compels a profound redefinition of B’s biological identity, from a chemically inert trace element to an ecological cofactor that sustains communication, cooperation, and structural stability within the host–microbiota ecosystem. Rather than acting as a mere biochemical participant in isolated enzymatic pathways, B—particularly in its microbiota-accessible complex forms (MABCs)—emerges as a symbiotic catalyst: a molecular mediator that harmonizes microbial quorum signaling, mucosal architecture, and host metabolic tone.

This reconceptualization aligns with a broader transformation in nutritional science: the shift from biochemical sufficiency to symbiotic functionality. In classical nutrition, essentiality was defined by the prevention of overt deficiency symptoms at the organismal level. In contrast, the MABC framework introduces a new dimension of ecological essentiality, where the maintenance of microbial homeostasis and barrier integrity becomes the foundation of long-term systemic health.

Consistent, low-level dietary exposure to MABCs appears indispensable for microbial eubiosis, mucosal resilience, and the moderation of immune–metabolic signaling networks. Conversely, insufficient colonic delivery of these complexes has been repeatedly linked to DYS, barrier fragility, increased permeability, and chronic low-grade inflammation, processes that collectively accelerate biological aging and functional decline [3,6,8]. The recurring pattern across experimental and ecological studies suggests that B’s biological impact depends not only on intake quantity but, crucially, on its access form and site of bioavailability.

#### 7.8.1. The Next Frontier: Integration Across Analytical, Clinical, and Regulatory Domains

Future progress requires validated workflows capable of distinguishing and quantifying MAB species with precision. High-resolution speciation analytics (UHPLC–MS, NMR, inductively coupled plasma–mass spectrometry (ICP–MS) with isotopic discrimination) and standardized AI-2B bioassays must become cornerstones of experimental design. Establishing international reference materials and interlaboratory calibration standards will be critical for reproducibility and global harmonization.

Translational studies must advance from correlational observations to form-controlled clinical trials, where matched elemental B doses are administered as PAB and MABC formulations. The decisive outcome is rescuing specificity—demonstrating that MABCs, but not systemically absorbed PAB forms, restore eubiosis, strengthen mucosal barriers, and attenuate inflammatory tone. Trials should incorporate multi-omics endpoints—metagenomics, metabolomics, and mucin proteomics—to capture the full spectrum of host–microbe interactions.

Conventional toxicological frameworks, which apply universal upper limits irrespective of chemical form or site of action, are inadequate for MABCs. A new form- and site-specific regulatory architecture should acknowledge the safety and functional relevance of colonic B exposure. Such frameworks would distinguish PAB from MABC forms, aligning labeling and intake guidelines with mechanistic evidence rather than generalized systemic thresholds.

#### 7.8.2. Nutritional Innovation

The evolution of dietary science toward microbiota-targeted nutrition will depend on developing foods and supplements optimized for colonic bioavailability. Metrics such as BND_T_ and BND_MA_ provide practical tools for evaluating diets and formulations. Future dietary recommendations may shift from “total B intake” toward “colonic delivery efficiency”, emphasizing B’s accessibility to the symbiotic microbiota as a determinant of healthspan.

When these domains—analytical precision, clinical validation, regulatory reform, and nutritional innovation—converge, B will stand not merely as a prebiotic trace element, but as a conditionally essential mediator of symbiosis. In this role, MABCs embody the next generation of nutritional science: bridging chemistry, microbiology, and physiology into a coherent model of ecological essentiality.

Through this integrative perspective, B’s role extends far beyond the prevention of deficiency—it becomes a catalyst of cooperation across biological systems, sustaining the equilibrium that defines health throughout the human lifespan.

## 8. Conclusions and Future Perspectives: Redefining Boron’s Nutritional Role Through the Microbiome

A convergent body of evidence reframes B from a marginal trace element to a site-specific, microbiome-dependent micronutrient whose physiological impact is governed by biological access form. We distinguish PAB—small, digestible inorganic/organic species absorbed in the upper gut that exert predominantly systemic and often transient actions—from MABCs—indigestible borate–diol conjugates that traverse the upper GI tract and act locally at the microbiome–mucus interface. Mechanistic and translational signals consistently indicate that MABCs sustain three interlocking functions: stabilization of interspecies QS via AI-2B, reinforcement of the mucin hydrogel through reversible borate–diol crosslinks, and a shift toward SCFA-centered ecology. These proximal effects propagate to permeability, immune tone, and multiple gut–organ axes.

Within this framework, recognition of B’s essentiality should be evaluated at the level of access form. MABCs meet classical essentiality criteria in a site-specific sense: (i) they are required to maintain symbiosis-dependent functions; (ii) deficiency phenotypes are coherent and testable (reduced AI-2B capacity, mucus thinning/penetration, DYS, low-grade inflammation); (iii) human cells do not synthesize MABCs (dietary exclusivity); (iv) chemical irreplaceability holds at key steps (AI-2B formation, borate–diol crosslinking); (v) they participate in identifiable functional structures (AI-2B, borate–mucin complexes); and (vi) dose–response signals are emerging for colonic delivery and readouts. By contrast, PAB aligns with non-essential, modulatory or pharmacological actions at typical intakes.

Sex-specific physiology, including differences in B handling, microbiome baselines, and hormone–microbiome crosstalk, likely modulates effect sizes and should be prospectively incorporated into study design and guidance. Operationally, BND metrics (BND_T_/BND_MA_) and speciation-guided assays provide a practical scaffold for dietary planning and product development aimed at colonic bioaccessibility, not merely systemic exposure.

Translational implications follow directly. In clinical nutrition, MABC-oriented patterns (pectin- and polyphenol-rich foods, gentle processing that preserves *cis*-diol scaffolds, and—where appropriate—co-formulation with fibers or probiotics) target barrier integrity and symbiotic signaling, with relevance to inflammaging, barrier-fragility syndromes, and DYS-linked conditions. In regulation and labeling, differentiating PAB from MABCs is necessary for risk assessment, intake recommendations, and on-pack communication (e.g., BND_MA_ content, colonic delivery factor). For evidence generation, access-form controlled randomized trials with colon-centric endpoints (fecal B/speciation, AI-2/AI-2B capacity, mucus integrity/penetration indices, permeability, SCFAs, inflammatory proxies), sex-stratification, and matrix-matched controls are decisive to adjudicate essentiality and to quantify intake thresholds. Analytical priorities include validated workflows for speciation stability and AI-2B quantification, with pre-registered pipelines to enhance reproducibility.

Taken together, these lines of evidence support a microbiota-aware redefinition of essential nutrients, in which MABCs exemplify a selective, structure-specific, site-dependent class of prebiotics. As analytical, clinical, and regulatory frameworks mature, B—in its MAFs—should be regarded as conditionally essential for maintaining HMS and, by extension, systemic resilience across the lifespan. In short: for B, form defines function; in the colon, function defines essentiality.

## Figures and Tables

**Figure 1 biomolecules-15-01711-f001:**
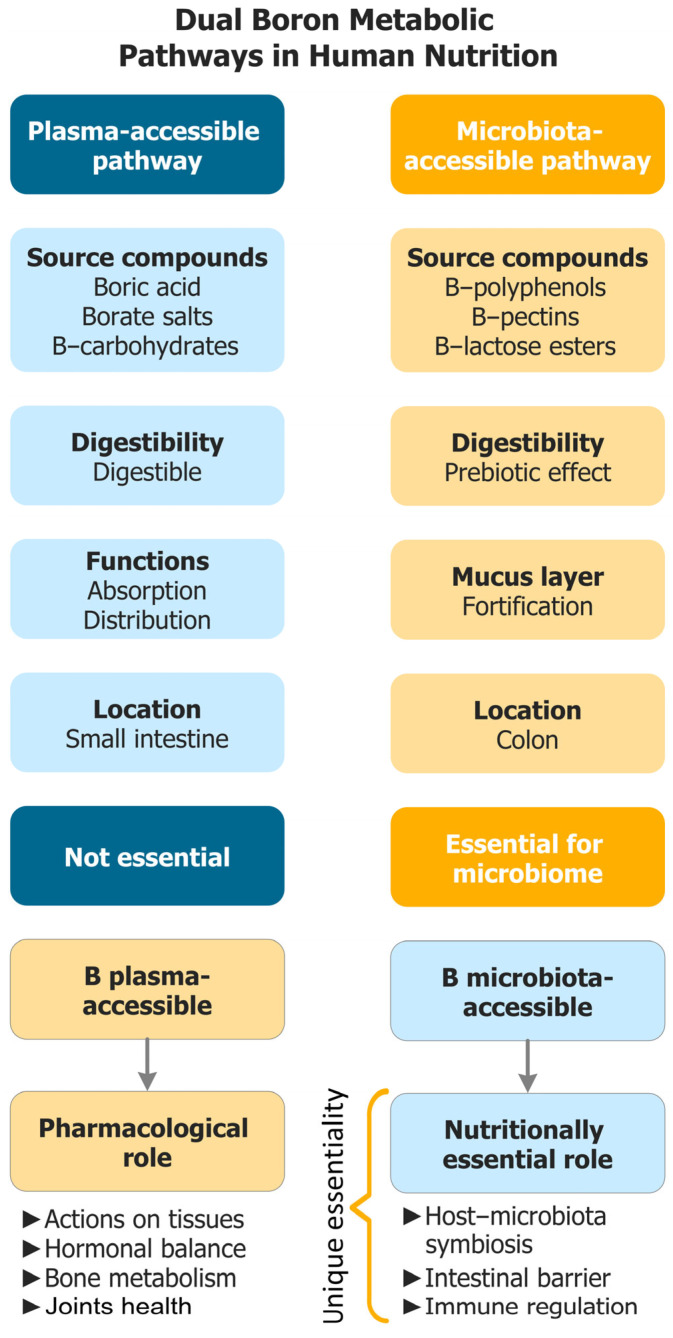
Dual-access model of B bioavailability. PAB is absorbed in the upper gastrointestinal tract and acts systemically and transiently, whereas MABCs resist digestion, reach the colon, and act locally at the microbiome–mucus interface (AI-2B signaling, mucin crosslinking, SCFA-centered ecology). The figure is a conceptual roadmap rather than a fully validated mechanistic model. B: Boron; AI-2B: Autoinducer-2–borate; MABCs: Microbiota-accessible boron complexes; PAB: Plasma-accessible boron; SCFA: Short-chain fatty acid.

**Figure 2 biomolecules-15-01711-f002:**
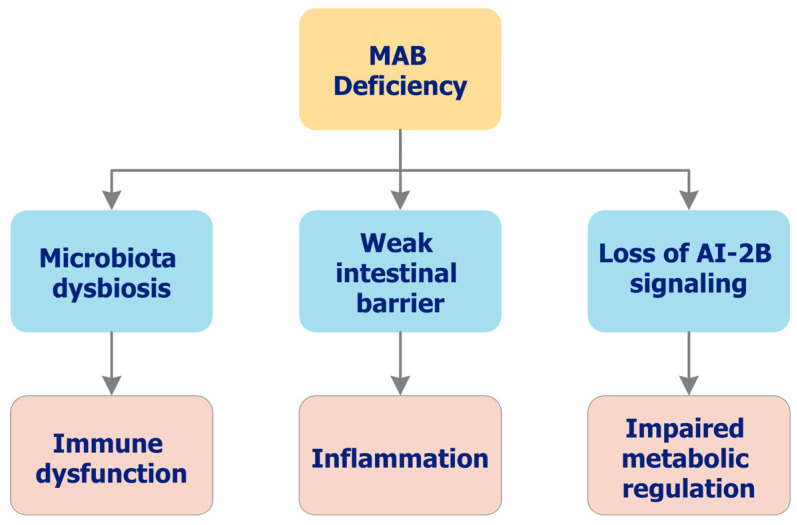
Functional consequences of inadequate MAB delivery (deficiency phenotype). Conceptual cascade linking low colonic availability of MAB to reduced AI-2B capacity, impaired mucus integrity, DYS, increased permeability, and low-grade inflammation. Downstream effects map along gut–organ axes (e.g., gut–joint, gut–brain), with symptom composites and inflammatory proxies increasing as barrier stewardship declines. The figure summarizes current concepts and is intended as a mechanistic framework rather than a fully validated clinical model. AI-2B: Autoinducer-2–borate; DYS: Dysbiosis; MAB: Microbiota-accessible boron.

**Figure 3 biomolecules-15-01711-f003:**
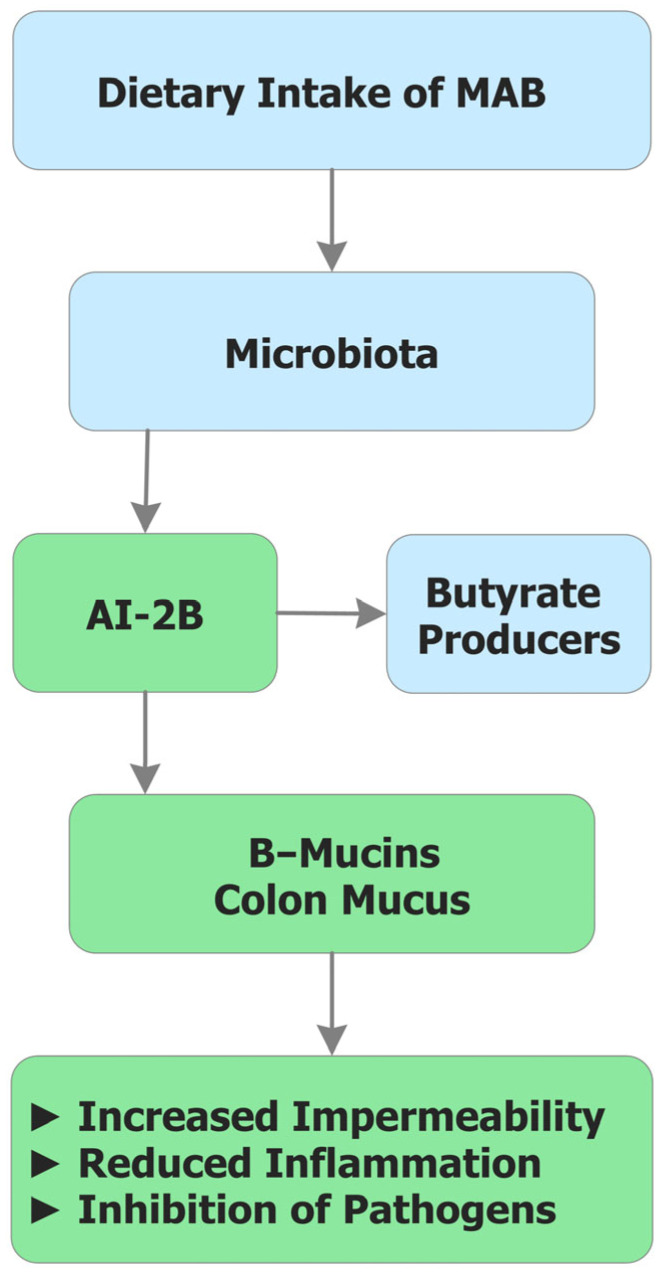
Integrated mechanism of MAB in the colon. Schematic overview of form-dependent mechanisms: (i) stabilization of AI-2B, supporting cross-species quorum behaviors; (ii) reversible borate–diol crosslinking that strengthens the mucin hydrogel and reduces bacterial encroachment; and (iii) a shift toward SCFA-centered metabolic ecology (notably BUT), with downstream modulation of permeability and immune tone. PAB provides transient systemic exposure with limited lumen access, whereas MAB ensures sustained, local availability at the microbiome–mucus interface. AI-2B: Autoinducer-2–borate; BUT: Butyrate; MAB: Microbiota-accessible boron; PAB: Plasma-accessible boron; SCFAs: Short-chain fatty acid.

**Figure 4 biomolecules-15-01711-f004:**
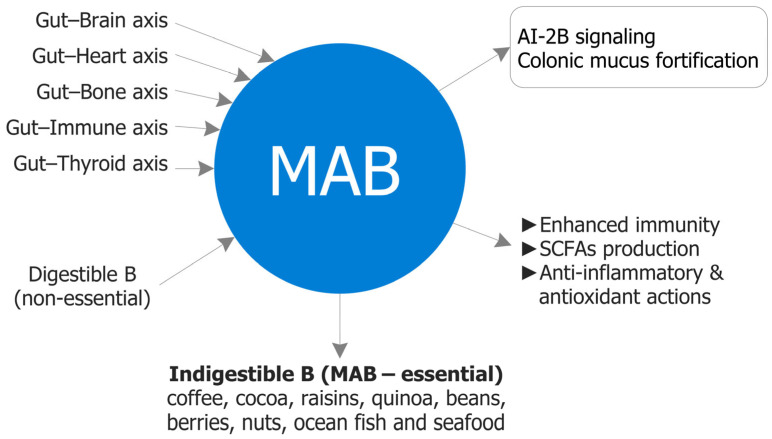
Strategic role of MAB. Conceptual map of form-dependent, colon-localized actions: (i) stabilization of AI-2 as AI-2B supporting cross-species quorum behaviors; (ii) reversible borate–diol crosslinking that strengthens the mucin hydrogel and reduces bacterial encroachment; and (iii) a shift toward SCFA-centered metabolic ecology with downstream modulation of permeability and immune tone. Systemic PAB exposure is transient and not a reliable substitute for lumen–mucus access. AI-2: Autoinducer-2; AI-2B: Autoinducer-2–borate; MAB: Microbiota-accessible boron; SCFAs: Short-chain fatty acids.

**Table 1 biomolecules-15-01711-t001:** Operational discriminators of access form (PAB vs. MABCs).

Domain	PAB	MABCs	Primary Biomarkers
GI fate	rapid small-intestinal absorption; minimal colonic arrival	upper-gut resistance; colonic arrival/retention	plasma/urine B (PAB); fecal B/speciation (MABCs)
site of action	systemic, transient	colon-localized, sustained	mucus thickness/penetration; bacterial encroachment; SCFAs
expected function	short-term systemic modulation	prebiotic/cofactor roles at the microbiome–mucus interface	proxies of quorum capacity (AI-2/AI-2B); barrier integrity
study design cue	pharmacokinetics/clearance; systemic endpoints	colonic delivery/readouts; access form matched trials	matched elemental B, distinct access forms

AI-2: Autoinducer-2; AI-2B: AI-2–borate; B: Boron; GI: Gastrointestinal; MABCs: Microbiota-accessible boron complexes; PAB: Plasma-accessible boron; SCFAs: Short-chain fatty acids.

**Table 2 biomolecules-15-01711-t002:** Mechanisms, biomarkers, and evidence base for MABCs vs. PAB.

Mechanism Domain	Primary Site/Action	Key Biomarkers/Readouts	Access-Form Expectation	Representative Evidence Base
AI-2B stabilization (QS)	lumen–mucus interface; cross-species coordination (resource sharing, biofilm structure, colonization resistance)	fecal AI-2 and AI-2B bioassays; community stability (α/β-diversity); resistance to opportunistic blooms under stress conditions	MABCs greater than PAB at matched elemental B	in vitro reporter systems; mixed-culture assays; animal models; emerging human data
mucin crosslinking and barrier reinforcement (borate–diol interactions)	mucus hydrogel; reversible borate–diol crosslinks tighten the polymer network	mucus thickness; penetration index (dextran/bead assays); bacteria–epithelium distance; intestinal permeability (e.g., FITC–dextran, L/M ratio); TJ transcripts; fecal B/speciation consistent with complexed forms	MABCs greater than PAB	in vitro rheology/microscopy; animal histology and permeability; emerging human proxies
SCFA-centered metabolic ecology	community metabolism and host GPCR signaling; colonocyte energetics	total SCFAs and BUT (GC/NMR); enrichment of butyrogenic taxa (e.g., *Faecalibacterium*, *Roseburia*); fecal calprotectin lower; systemic hs-CRP/IL-6 lower in parallel with barrier gains	MABCs equal to or greater than PAB, often greater with sustained delivery	in vitro fermentations; animal models; emerging human cohorts/interventions
reverse B trapping (plasma ↔ colon recycling)	bidirectional exchange; trapping as borate esters in diol-rich matrices (mucin O-glycans, phenolics, microbial EPS)	fecal B/speciation during sustained intake; coupling with mucus metrics and AI-2B capacity; attenuation when diol substrates are limited	requires diol-rich matrices (MABC context), not reliably achieved by PAB alone	animal kinetic studies; emerging human observations

AI-2: Autoinducer-2; AI-2B: Autoinducer-2–borate; B: Boron; BUT: Butyrate; EPS: Exopolysaccharides; FITC: Fluorescein isothiocyanate; GC: Gas chromatography; GPCR: G protein-coupled receptor; hs-CRP: High-sensitivity C-reactive protein; IL-6: Interleukin-6; L/M: Lactulose/mannitol; MABCs: Microbiota-accessible boron complexes; NMR: Nuclear magnetic resonance; PAB: Plasma-accessible boron; QS: Quorum sensing; SCFA: Short-chain fatty acid; TJ: Tight junction.

**Table 3 biomolecules-15-01711-t003:** Sex-specific B–microbiota interactions.

Aspect	Female	Male
hair B range (μg/g)	0.472–3.89	0.771–6.51
whole-blood B	similar to male	similar to female
DYS-linked conditions	higher prevalence: osteoporosis, PCOS, ovarian cancer, IBD	lower prevalence; higher metabolic syndrome risk
AI-2B biomarker utility	potential sex-sensitive indicator of MAB status	potential indicator; less studied
predominant aging-cohort taxa	*Prevotella copri*, *Eubacterium rectale*, *Roseburia inulinivorans*	*Lactobacillus gasseri*, *L. oris*, *Clostridium hathewayi*, *Eggerthella lenta*
estrogen–GM modulation	strong; probiotic modulation documented	less impacted; mainly indirect effects
plasma B vs. mortality	positive association with all-cause mortality	no significant association detected

AI-2B: Autoinducer-2–borate; B: Boron; DYS: Dysbiosis; GM: Gut microbiota; IBD: Inflammatory bowel disease; MAB: Microbiota-accessible boron; PCOS: Polycystic ovary syndrome.

**Table 4 biomolecules-15-01711-t004:** BND and corresponding biomarker correlations.

Parameter/Domain	Descriptor	Expected Correlation with MABC Intake	Primary Biomarkers/Readouts
BND_T_ vs. BND_MA_	ratio of total to indigestible B per 1000 kcal	↑ BND_MA_ correlates with improved colonic delivery	fecal B concentration; B speciation profile
QS signaling	AI-2/AI-2B formation and interspecies coordination	↑ AI-2B capacity with higher MABC delivery	fecal AI-2/AI-2B bioassays
mucus-barrier integrity	borate–diol crosslinking in O-glycosylated mucins	↑ mucus thickness; ↓ penetration index	mucus thickness, dextran penetration, bacterial distance
gut permeability and inflammation	TJ regulation and immune tone	↓ permeability; ↓ calprotectin, hs-CRP, IL-6	FITC-dextran, L/M ratio, fecal and serum inflammatory markers
metabolic ecology	SCFA production and butyrogenic taxa	↑ total SCFAs (mainly BUT) and ↑ *Roseburia* spp., *Faecalibacterium* spp.	SCFAs (GC/NMR); 16S diversity indices

↑: Increase; ↓: Decrease; AI-2: Autoinducer-2; AI-2B: Autoinducer-2–borate; B: Boron; BND: Boron nutrient density; BND_MA_: Microbiota-accessible boron nutrient density; BND_T_: Total boron nutrient density; BUT: Butyrate; FITC: Fluorescein isothiocyanate; GC: Gas chromatography; hs-CRP: High-sensitivity C-reactive protein; IL-6: Interleukin-6; L/M: Lactulose/mannitol; MABC: Microbiota-accessible boron complex; NMR: Nuclear magnetic resonance; QS: Quorum sensing; SCFAs: Short-chain fatty acids; TJ: Tight junction.

**Table 5 biomolecules-15-01711-t005:** Intake targets and candidate biomarkers for MABC delivery.

Metric	Definition/Assay	Physiological Rationale	Target/Interpretation
BND_T_	mg/day (dietary analysis)	total dietary B pool, of which ~10% reaches colon as BND_MA_	around 10 mg/day corresponding to at least 1 mg/day MABCs
MABCs delivered	mg/day (estimated from dietary matrix or formulation)	reflects the form-dependent colonic action of MAB	at least 1 mg/day achievable through plant-based diets
fecal B/speciation	mg/day; speciation profile (complexed vs. free B species)	serves as a proxy for colonic delivery and retention of B complexes	approximately 1 mg/day with predominance of complexed species
AI-2/AI-2B (fecal)	relative units (orthogonal assays)	represents QS activity associated with microbial eubiosis	increase compared with baseline after intervention
mucus thickness/ penetration	histological or imaging indices; bacterial encroachment distance	indicates barrier integrity mediated by borate–diol crosslinking	improvement relative to baseline
permeability	L/M ratio; zonulin or equivalent markers	reflects epithelial barrier function	reduction toward physiological values
SCFAs (acetate, propionate, BUT)	mmol/kg (stool; GC or NMR methods)	marker of microbial metabolism and colonocyte energetics	increased BUT consistent with eubiosis

AI-2: Autoinducer-2; AI-2B: Autoinducer-2–borate; B: Boron; BND_MA_: Microbiota-accessible boron nutrient density; BND_T_: Total boron nutrient density; BUT: Butyrate; GC: Gas chromatography; L/M: Lactulose/mannitol; MAB: Microbiota-accessible boron; MABCs: Microbiota-accessible boron complexes; NMR: Nuclear magnetic resonance; SCFA: Short-chain fatty acid.

## Data Availability

The original contributions presented in this study are included in the article. Further inquiries can be directed to the corresponding author.

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
