# Peer review of "Boron Bioavailability Revisited: From Plasma-Accessible Species to Microbiota-Accessible Complexes—Implications for Nutritional Essentiality"

_biomolecules, 2025, doi:10.3390/biom15121711_

Round 1

Reviewer 1 Report

Comments and Suggestions for Authors

The entire manuscript has an unclear structure and flow of information, with concepts repeated throughout the manuscript, and no tables are provided to allow for easy and quick location of the results of the studies to which the authors are referring.

Abstract section is confusing and does not describes and introduction, materials and methods, results and conclusions. It must be rewritten.

The entire manuscript has an unclear structure and flow of information, with concepts repeated throughout the manuscript, and no tables are provided to allow for easy and quick location of the results of the studies to which the authors are referring.

The conclusions are two full pages long, listing a large amount of different data, often without any obvious connection between them, and without reaching a conclusion as such, but rather a kind of summary of findings.

There are large sections of text in which the relationship with boron is more than debatable, and which could have been included in any article discussing the effects of any other mineral element. For example, on page 4, only lines 174-178 have any connection with boron. On page 5, lines 199-223, the same applies.

In general, the manuscript suffers from having a linear structure in which a specific problem or aspect to be investigated is cited, the methods used to determine the solution to a problem are described, the results obtained by other authors are cited in a clear and concise manner, and finally a conclusion is established, even including aspects of the authors' personal opinion. In this manuscript, a circular structure is established, in which the authors combine general issues with others that are more related to boron, finally arriving at a series of generalities in the conclusions.

Author Response

Dear Reviewer,

First of all, we would like to address you many thanks for your accurate observations and valuable comments. We used all these and improved the paper accordingly.

All changes in the revised manuscript were highlighted on a yellow background.

The following changes have been made to the Manuscript (ID: biomolecules-3937117):

Reviewer #1 questions/comments

Comments 1:

Abstract section is confusing and does not describes and introduction, materials and methods, results and conclusions. It must be rewritten.

Response 1:

Thank you very much for your helpful suggestion. “Abstract” section has been completely restructured according to the reviewer’s recommendations. It now follows the Journal’s four-part format (Background, Methods, Results, and Conclusions), but without headings, clearly summarizes the dual-pathway framework (PAB vs. MABCs), highlights the main mechanistic findings (AI-2B signaling, mucin reinforcement), and ends with a concise, outcome-oriented conclusion on B’s microbiota-dependent essentiality. Also, “Keywords” section has been revised accordingly. (See page 1, lines 42–50; page 2, lines 51–61).

Comments 2:

The entire manuscript has an unclear structure and flow of information, with concepts repeated throughout the manuscript, and no tables are provided to allow for easy and quick location of the results of the studies to which the authors are referring.

Response 2:

Thank you very much for your valuable feedback on manuscript structure and flow of information. “1. Introduction” section has been fully reorganized into clearly defined subsections (1.1 to 1.5) to enhance clarity, linearity, and readability. Redundant descriptions of plasma-accessible boron (PAB) have been removed, while the new “1.4. Aim and Scope” paragraph explicitly states the objectives and scientific rationale of the review. To further improve navigability, a dedicated subsection, “1.5. Structure of This Review”, has been added at the end of the “1. Introduction” section. This paragraph provides a concise overview of the manuscript’s organization, guiding the reader from the conceptual framework (Section 2) to mechanistic evidence (Section 3), experimental and clinical validation (Section 4), and nutritional–translational implications (Sections 5 and 6), culminating in key conclusions (Section 7). These revisions directly address the reviewer’s concerns about unclear flow and repetition, providing a logical progression from chemistry to physiology and application while preserving the original scientific content and conceptual integrity of the paper. (See “1. Introduction” section).

Section 2 has been restructured to resolve circularity and improve linear flow. A new “2.1 Why Essentiality Must be Evaluated by Access Form” subsection bridges classical essentiality to our dual-access framework (PAB vs. MABCs) and is supported by Figure 1 (dual-access model). Subsections 2.2 and 2.3 were streamlined and anchored explicitly in B biology. Subsection 2.4 clarifies when B is essential (site-specific, microbiota-accessible delivery), while Table 1 provides operational discriminators (gastrointestinal fate, site of action, expected functions, biomarkers) to guide study design. Two new subsections have been added: “2.5. Mechanistic Preview—Why Form Determines Function (PAB vs. MABCs)” and “2.6. MABCs and Nutritional Essentiality—Conceptual Position and Testing Strategy”. (See “2. The Concept of Nutritional Essentiality” section).

Section 5 has been reworked to foreground access-form essentiality and to provide a roadmap of open needs. Subsection 5.1 formalizes essentiality at the site-specific (colonic) level, with a defined deficiency phenotype and MABC-specific rescue criteria. Subsections 5.2 and 5.3 specify standardized analytics (fecal B/speciation; AI-2/AI-2B; barrier imaging/permeability) and trial designs that compare matched elemental doses of PAB vs. MABCs on colonic endpoints. Practical targets and discriminative biomarkers have been consolidated in Table 5 and kept clinical detail minimal here, deferring outcome syntheses to later sections. Safety, regulatory, and food-design considerations (subsections 5.4 to 5.6) have been reframed to distinguish MABCs from PAB in exposure and mechanism. Figure 4 serves as a forward-looking concept map rather than mechanistic proof, addressing the request for clarity without redundancy. (See “5. Challenges and Future Directions for Recognizing Prebiotic MABCs as Nutritionally Essential” section).

We thank the reviewer again for this constructive and insightful recommendation. Section 6 has been completely restructured and expanded to present a more academically integrated and mechanistically coherent discussion. The revision replaces the earlier schematic layout with a continuous, narrative style appropriate for a high-level scientific review. It now situates MABCs within a scientifically grounded framework for clinical nutrition, safety regulation, and innovation, consistent with contemporary expectations for a major review in this field. (See “6. Clinical and Regulatory Implications” section).

Section 6 closes the translational loop—from mechanistic understanding to clinical testing, dietary modeling, safety evaluation, and regulatory classification—forming a cohesive framework for future implementation of MABCs in human health strategies:

▪ 6.1. – Clinical nutrition now frames MABCs as a therapeutic platform targeting barrier fragility, inflammaging, and dysbiosis-linked disorders across gut–organ axes, with detailed mechanistic interpretation and clinically relevant readouts;

▪ 6.2. – Safety and toxicology develop an access form aware safety framework distinguishing PAB (systemic, rapidly cleared) from MABCs (indigestible, lumen-acting), integrating toxicokinetics, sex-specific considerations, and assay discipline;

▪ 6.3. – Dietary guidance and labeling links total boron nutrient density (BNDT) and microbiota-accessible boron nutrient density (BNDMA) to colonic availability, providing measurable targets for intake, adherence, and effect biomarkers;

▪ 6.4. – Regulatory pathways articulate a stepwise dossier approach (chemistry, exposure, mechanism, safety, and substantiation) advocating for the recognition of form- and site-specific essentiality within prebiotic frameworks;

▪ 6.5. – Food design and medical nutrition translate theory into technological innovation, emphasizing protection of boron–diol conjugates through the upper gut, preservation of functional scaffolds, speciation-guided quality control, and mechanistic co-formulation;

▪ 6.6. – Trial design and causality provide a clinically actionable template for proving causality, specifying colonic endpoints (fecal B/speciation, AI-2B, SCFAs, permeability indices) and methodological rigor (sex-stratified, baseline microbiome-stratified, pre-registered pipelines);

▪ 6.7. – Methodological pitfalls and confounders now elaborate on pH-sensitive speciation artifacts, co-nutrient interference, ecological variability of AI-2B signaling, and translational scaling between species, offering detailed corrective measures;

▪ 6.8. – Outlook expands into a forward-looking synthesis: B as an ecological cofactor and symbiotic catalyst, integrating analytical, clinical, regulatory, and nutritional dimensions of future MABC research. (See 6.1 to 6.8 expanded subsections).

Comments 3:

The conclusions are two full pages long, listing a large amount of different data, often without any obvious connection between them, and without reaching a conclusion as such, but rather a kind of summary of findings.

Response 3:

Thank you for pointing this out. The revised Conclusions and Future Perspectives section now provides a rigorous and integrative synthesis, connecting mechanistic insight with practical translation. It addresses all reviewer concerns by clarifying B’s dual-access framework, aligning the argument with essentiality theory, and projecting a clear trajectory for future research, dietary implementation, and regulatory recognition of MABCs as a new paradigm in nutritional science. (See “7. Conclusions and Future Perspectives: Redefining Boron’s Nutritional Role Through the Microbiome” section).

Comments 4:

There are large sections of text in which the relationship with boron is more than debatable, and which could have been included in any article discussing the effects of any other mineral element. For example, on page 4, only lines 174-178 have any connection with boron. On page 5, lines 199-223, the same applies.

Response 4:

We thank the Reviewer for the insightful comments that helped us refine both the mechanistic clarity and the physiological framing of Section 3. We have revised this section substantially to strengthen the logical flow, integrate recent references, and align with the Reviewer’s emphasis on mechanistic evidence and essentiality criteria. These revisions clarify that MABCs meet recognized essentiality criteria through site-specific, non-redundant mechanisms that sustain host–microbiota symbiosis. (See “3. Essentiality of MABCs for Normal Human Physiology” section).

The revised Section 3 thus connects chemical accessibility, physiological outcomes, and nutritional essentiality in a coherent, testable framework:

▪ Conceptual structure. The section now begins with a concise definition of MABCs and a stepwise framework linking molecular chemistry to host physiology. The revised introduction to Section 3 emphasizes that essentiality depends on form-dependent accessibility rather than on B’s intrinsic presence alone.

▪ Mechanistic subsections (3.1 to 3.5). Each mechanism (AI-2B QS, mucin crosslinking, SCFA-centered ecology, reverse B trapping, and dietary dependence) is now presented with a short mechanistic description, corresponding biomarkers/predictions, and an essentiality link, indicating how MABCs uniquely fulfill a physiological role that cannot be substituted by plasma-accessible boron (PAB).

▪ Integration of visual elements. Figure 2 (integrated mechanism of MABCs in the colon) summarizes the three primary domains: AI-2B formation, mucin reinforcement, and SCFA-centered ecology. Figure 3 (deficiency phenotype) depicts the functional consequences of inadequate MABC delivery across gut–organ axes. Table 2 consolidates all mechanistic domains, biomarkers, and evidence levels into a standardized scientific format, replacing the earlier heuristic table.

▪ Evidence base and dose–response logic. Subsection 3.6 incorporates animal and human evidence supporting an intake–response relationship for fecal B, mucus barrier integrity, and AI-2B signaling.

▪ Terminological and stylistic adjustments. Redundant or speculative phrasing was removed. Terminology (AI-2B, SCFA, MABC, PAB, HMS) was standardized across all subsections for clarity and consistency. (See “3. Essentiality of MABCs for Normal Human Physiology” section).

Comments 5:

In general, the manuscript suffers from having a linear structure in which a specific problem or aspect to be investigated is cited, the methods used to determine the solution to a problem are described, the results obtained by other authors are cited in a clear and concise manner, and finally a conclusion is established, even including aspects of the authors’ personal opinion. In this manuscript, a circular structure is established, in which the authors combine general issues with others that are more related to boron, finally arriving at a series of generalities in the conclusions.

Response 5:

We thank the reviewer for highlighting the need to clarify the physiological consequences of low MABC intake and to better differentiate the systemic (PAB) vs. colonic (MABC) pathways of B utilization. Section 4 has been comprehensively revised to improve conceptual flow, mechanistic precision, and linkage to measurable biomarkers:

▪ A concise introduction now restates the dual-access model (PAB vs. MABCs) and anchors physiological outcomes to site-specific accessibility.

▪ The revised subsection 4.1 integrates the expected biological effects of low MABC intake with mechanistic and biomarker-level predictions (covering AI-2B signaling capacity, mucus barrier metrics, permeability, and inflammatory proxies), clarifying how deficiency at the colonic interface manifests systemically.

▪ Subsection 4.2 has been revised to address sex-specific physiological responses, incorporating evidence on B kinetics, gut microbiota composition, and hormonal modulation (β-glucuronidase-mediated estrogen recycling). The updated Table 3 now summarizes comparative data for males and females, identifying potential sex-sensitive biomarkers (notably fecal AI-2B).

▪ Subsection 4.3 introduces a refined framework for boron nutrient density (BND), distinguishing total (BNDT) from microbiota-accessible (BNDMA) fractions. Quantitative relationships (e.g., ≥ 10 mg/day BNDT → ≥ 1 mg/day MABCs) are provided, along with feasible intake markers (fecal B ≥ 1 mg/day). This subsection also highlights the dual structural and signaling functions of MABCs and situates them within a symbiotic evolutionary context. (See “4. Physiological Response During Very Low Intake of Boron Complexes” section)

Authors very much appreciated the encouraging, critical, and constructive comments on this manuscript by the Reviewer. The comments have been very thorough and useful in improving the manuscript.

We would like to thank the Reviewer again for taking the time to review our manuscript.

Academic Editor questions/comments

Comments 1:

During the technical check of your manuscript, we noticed that a high proportion of the cited references belong to you or your co-authors: Refs. 3, 5, 6, 8, 10, 13, 31, 7, 54, 56, 57, which is a self-citation rate of about 15.49%.

As MDPI is a member of COPE (https://publicationethics.org/), all references in our published articles must contribute to the scholarly content of the paper and avoid bias (self-citations, journal citations, school of thought, etc.) and reflect the current state of knowledge in the field. We encourage you to consider this and reduce the self-citations to make sure that only the most relevant citations are kept. If you feel that the citations to the previous works are essential, an Academic Editor will check the appropriateness of these citations. If excessive self-citations are found at this stage your manuscript may be rejected.

Response 1:

Thank you very much for your observation. References [6] (Biţă et al., 2023), [7] (Scorei et al., 2023), [31] (Donoiu et al., 2018), [54] (Scorei et al., 2025), [56] (Soriano-Ursúa et al., 2014) and [57] (Barrón-González et al., 2023), numbered according to the initial submission, have been removed from the manuscript and from the citation list. We systematically reviewed references and reduced self-citations rate to 7.69%.

We have also introduced other additions/modifications that we hope will improve the quality of the manuscript:

▪ Two e-mail addresses have been updated: “laura.dinca@umfcv.ro” instead of “eu-office@umfcv.ro” (for L.D.); “iurie.pinzaru@ansp.gov.md” instead of “iurie.panzaru@ansp.gov.md” (for I.P.). (See page 1, lines 23 & 27).

▪ Tables and Figures have been renumbered accordingly.

▪ The Reference list has been entirely checked and renumbered accordingly.

▪ All abbreviations have been defined the first time they appear in the text.

▪ Some grammar, stylistic or spelling errors have been corrected.

Kind regards,

Ion Romulus SCOREI, Professor, PhD

Reviewer 2 Report

Comments and Suggestions for Authors

The importance of boron in the lives of plants and animals is becoming increasingly clear. This review needs to include more information specifically about its role in the many development cycles of plants and living organisms, as well as what happens when there is a deficiency and/or excess of this element.

The size of the electronic journal allows for a significant expansion of the list of cited scientific literature (articles in scientific journals, many of which are available for review). It is necessary to expand the information - to describe in detail all possible ways for living organisms to obtain boron, and how the deficiency and excess of this element affects living organisms.

Authors, please note the number of co-authors in this article. It's too high. Did everyone actually participate in data collection and processing?

Comments on the Quality of English Language

All articles by authors whose native language is not English must be reviewed by a native English speaker to correct any subtleties in the presentation and reporting of the authors' findings.

Author Response

Dear Reviewer,

First of all, we would like to address you many thanks for your accurate observations and valuable comments. We used all these and improved the paper accordingly.

All changes in the revised manuscript were highlighted on a yellow background.

The following changes have been made to the Manuscript (ID: biomolecules-3937117):

Reviewer #2 questions/comments

Comments 1:

The importance of boron in the lives of plants and animals is becoming increasingly clear. This review needs to include more information specifically about its role in the many development cycles of plants and living organisms, as well as what happens when there is a deficiency and/or excess of this element.

Response 1:

Thank you very much for your helpful suggestion. The manuscript has been revised accordingly. (See page 7, lines 277–283; page 22, lines 855–862).

Comments 2:

The size of the electronic journal allows for a significant expansion of the list of cited scientific literature (articles in scientific journals, many of which are available for review). It is necessary to expand the information - to describe in detail all possible ways for living organisms to obtain boron, and how the deficiency and excess of this element affects living organisms.

Response 2:

Thank you very much for your insightful comment. Citations have been expanded and diversified throughout Sections 1–4 and 6, reducing self-citation (see Academic Editor response below). We also added a concise pathway map (dietary, water, matrix-bound MABCs; and the plasma ↔ colon ‘reverse trapping’ phenomenon as exchange, not endogenous synthesis) and summarized deficiency/excess consequences relevant to colon-level endpoints. (See Sections 1–4 and 6).

Comments 3:

Authors, please note the number of co-authors in this article. It’s too high. Did everyone actually participate in data collection and processing?

Response 3:

Thank you for pointing this out. Individual contribution of each author has been provided accordingly. Considering CRediT taxonomy (https://img.mdpi.org/data/contributor-role-instruction.pdf), authorship is consistent with journal policy. C.G.P., C.E.B., L.D., S.Ş., M.-V.R., I.P., C.F., D.-R.H., J.M.H., C.A.S., J.N. and D.I.G. were responsible for investigation (“Conducting a research and investigation process, specifically performing the experiments, or data/evidence collection”). (See “Author Contributions” section).

Comments 4:

Comments on the Quality of English Language: The English could be improved to more clearly express the research. All articles by authors whose native language is not English must be reviewed by a native English speaker to correct any subtleties in the presentation and reporting of the authors’ findings.

Response 4:

Thank you very much for your observation. The full manuscript was edited by a native-speaking English scientific editor to improve clarity, tense consistency, and hedging where appropriate.

Authors very much appreciated the encouraging, critical, and constructive comments on this manuscript by the Reviewer. The comments have been very thorough and useful in improving the manuscript.

We would like to thank the Reviewer again for taking the time to review our manuscript.

Academic Editor questions/comments

Comments 1:

During the technical check of your manuscript, we noticed that a high proportion of the cited references belong to you or your co-authors: Refs. 3, 5, 6, 8, 10, 13, 31, 7, 54, 56, 57, which is a self-citation rate of about 15.49%.

As MDPI is a member of COPE (https://publicationethics.org/), all references in our published articles must contribute to the scholarly content of the paper and avoid bias (self-citations, journal citations, school of thought, etc.) and reflect the current state of knowledge in the field. We encourage you to consider this and reduce the self-citations to make sure that only the most relevant citations are kept. If you feel that the citations to the previous works are essential, an Academic Editor will check the appropriateness of these citations. If excessive self-citations are found at this stage your manuscript may be rejected.

Response 1:

Thank you very much for your observation. References [6] (Biţă et al., 2023), [7] (Scorei et al., 2023), [31] (Donoiu et al., 2018), [54] (Scorei et al., 2025), [56] (Soriano-Ursúa et al., 2014) and [57] (Barrón-González et al., 2023), numbered according to the initial submission, have been removed from the manuscript and from the citation list. We systematically reviewed references and reduced self-citations rate to 7.69%.

We have also introduced other additions/modifications that we hope will improve the quality of the manuscript:

▪ Two e-mail addresses have been updated: “laura.dinca@umfcv.ro” instead of “eu-office@umfcv.ro” (for L.D.); “iurie.pinzaru@ansp.gov.md” instead of “iurie.panzaru@ansp.gov.md” (for I.P.). (See page 1, lines 23 & 27).

▪ Tables and Figures have been renumbered accordingly.

▪ The Reference list has been entirely checked and renumbered accordingly.

▪ All abbreviations have been defined the first time they appear in the text.

▪ Some grammar, stylistic or spelling errors have been corrected.

Kind regards,

Ion Romulus SCOREI, Professor, PhD

Reviewer 3 Report

Comments and Suggestions for Authors

The manuscript is well written, conceptually original, and successfully integrates principles from chemistry, microbiology, and nutrition into a coherent and compelling framework. The proposed dual‐pathway model distinguishing plasma‐accessible boron (PAB) from microbiota‐accessible boron complexes (MABCs) represents a significant conceptual advance for re‐evaluating boron’s nutritional essentiality through the lens of host–microbiota interactions. I recommend minor revision. Specifically, the authors should clarify which mechanistic aspects are supported by direct empirical evidence and which remain hypothetical, ensuring that speculative mechanisms are appropriately framed as emerging concepts. In addition, moderating the language in sections that currently imply established causality would further enhance readability, conceptual precision, and scientific rigor.

Author Response

Dear Reviewer,

First of all, we would like to address you many thanks for your accurate observations and valuable comments. We used all these and improved the paper accordingly.

All changes in the revised manuscript were highlighted on a yellow background.

The following changes have been made to the Manuscript (ID: biomolecules-3937117):

Reviewer #3 questions/comments

The manuscript is well written, conceptually original, and successfully integrates principles from chemistry, microbiology, and nutrition into a coherent and compelling framework. The proposed dual‐pathway model distinguishing plasma‐accessible boron (PAB) from microbiota‐accessible boron complexes (MABCs) represents a significant conceptual advance for re‐evaluating boron’s nutritional essentiality through the lens of host–microbiota interactions.

I recommend minor revision.

Comments 1:

Specifically, the authors should clarify which mechanistic aspects are supported by direct empirical evidence and which remain hypothetical, ensuring that speculative mechanisms are appropriately framed as emerging concepts.

Response 1:

Thank you very much for your helpful suggestion. Each mechanistic domain has been labeled in Section 3, with evidence status in prose (in vitro / animal / human signals) and used neutral, non-causal language (“supports,” “is associated with”, “consistent with”, “testable prediction”). Table 2 summarizes discriminative biomarkers for MABCs vs. PAB at matched elemental B. (See “3. Essentiality of MABCs for Normal Human Physiology” section).

Comments 2:

In addition, moderating the language in sections that currently imply established causality would further enhance readability, conceptual precision, and scientific rigor.

Response 2:

Thank you very much for pointing this out. Throughout, deterministic claims have been replaced with graded phrasing and added “rescue specificity” as the decisive test (equal elemental boron as MABCs vs. PAB diverging on colon endpoints), explicitly framed as a prospective criterion.

Authors very much appreciated the encouraging, critical, and constructive comments on this manuscript by the Reviewer. The comments have been very thorough and useful in improving the manuscript.

We would like to thank the Reviewer again for taking the time to review our manuscript.

Academic Editor questions/comments

Comments 1:

During the technical check of your manuscript, we noticed that a high proportion of the cited references belong to you or your co-authors: Refs. 3, 5, 6, 8, 10, 13, 31, 7, 54, 56, 57, which is a self-citation rate of about 15.49%.

As MDPI is a member of COPE (https://publicationethics.org/), all references in our published articles must contribute to the scholarly content of the paper and avoid bias (self-citations, journal citations, school of thought, etc.) and reflect the current state of knowledge in the field. We encourage you to consider this and reduce the self-citations to make sure that only the most relevant citations are kept. If you feel that the citations to the previous works are essential, an Academic Editor will check the appropriateness of these citations. If excessive self-citations are found at this stage your manuscript may be rejected.

Response 1:

Thank you very much for your observation. References [6] (Biţă et al., 2023), [7] (Scorei et al., 2023), [31] (Donoiu et al., 2018), [54] (Scorei et al., 2025), [56] (Soriano-Ursúa et al., 2014) and [57] (Barrón-González et al., 2023), numbered according to the initial submission, have been removed from the manuscript and from the citation list. We systematically reviewed references and reduced self-citations rate to 7.69%.

We have also introduced other additions/modifications that we hope will improve the quality of the manuscript:

▪ Two e-mail addresses have been updated: “laura.dinca@umfcv.ro” instead of “eu-office@umfcv.ro” (for L.D.); “iurie.pinzaru@ansp.gov.md” instead of “iurie.panzaru@ansp.gov.md” (for I.P.). (See page 1, lines 23 & 27).

▪ Tables and Figures have been renumbered accordingly.

▪ The Reference list has been entirely checked and renumbered accordingly.

▪ All abbreviations have been defined the first time they appear in the text.

▪ Some grammar, stylistic or spelling errors have been corrected.

Kind regards,

Ion Romulus SCOREI, Professor, PhD

Round 2

Reviewer 1 Report

Comments and Suggestions for Authors

The authors have done a significant amount of work during the revision process, the text has improved substantially, and the resulting manuscript appears to be a completely different text. I believe it can now be considered for publication. I suggest removing those headings.

In any case, Subheadings 1.4 and 1.5 are not introduction, them are materials and methods.

Lines 155-164: No reference supports this text.

Further information about the search methodology and criteria for selecting and excluding articles, including adherence to PRISMA criteria and the search algorithm, is required.

Author Response

Dear Reviewer,

First of all, we would like to address you many thanks for your accurate observations and valuable comments. We used all these and improved the paper accordingly.

All changes in the revised manuscript were highlighted on a yellow background.

The following changes have been made to the Manuscript (ID: biomolecules-3937117):

Reviewer #1 questions/comments

The authors have done a significant amount of work during the revision process, the text has improved substantially, and the resulting manuscript appears to be a completely different text. I believe it can now be considered for publication.

Comments 1:

I suggest removing those headings. In any case, Subheadings 1.4 and 1.5 are not introduction, them are materials and methods.

Response 1:

Thank you very much for your helpful suggestion. The “1. Introduction” section has been reorganized, all subheadings (former Subsections 1.1–1.5) being removed as suggested. The Introduction is now presented as a continuous narrative that focuses solely on the conceptual background: historical context and classification of B, the distinction between PAB and MABCs, and the overall aims and structure of the review. (See “1. Introduction” section).

Comments 2:

Lines 155-164: No reference supports this text.

Response 2:

Thank you very much for your observation. Subsection 3.1 (formerly 2.1) has been rewritten to make its reasoning clearer and have anchored each key statement in the existing literature, using independent (non-self) sources on nutritional essentiality, micronutrient bioavailability, and microbiota-accessible nutrients. (See “3.1. Why Essentiality Must be Evaluated by Access Form” subsection).

Comments 3:

Further information about the search methodology and criteria for selecting and excluding articles, including adherence to PRISMA criteria and the search algorithm, is required.

Response 3:

Thank you very much for your insightful comment. We appreciate this important suggestion and have substantially revised the manuscript to provide a clearer and more transparent description of the search methodology. A new, fully developed Section 2 – “Search Strategy and Selection Criteria (PRISMA-Inspired Approach)” – has been added to clearly present our search methodology, inclusion/exclusion criteria, and PRISMA-aligned structure. We hope that this expanded methodological description satisfactorily meets the reviewer’s request. (See “2. Search Strategy and Selection Criteria (PRISMA-Inspired Approach)” section).

Authors very much appreciated the encouraging, critical, and constructive comments on this manuscript by the Reviewer. The comments have been very thorough and useful in improving the manuscript.

We would like to thank the Reviewer again for taking the time to review our manuscript.

We have also introduced other additions/modifications that we hope will improve the quality of the manuscript:

▪ All sections, subsections and sub-subsections have been renumbered accordingly.

▪ All abbreviations have been defined the first time they appear in the text.

▪ Eleven new citations have been added: [13] (Page et al., 2021); [21] (Hou et al., 2015); [22] (Sandström, 2001); [23] (Hambidge, 2010); [24] (Melse-Boonstra, 2020); [25] (Richards et al., 2025); [26] (Sonnenburg & Sonnenburg, 2014); [27] (Valdes et al., 2018); [28] (Bedu-Ferrari et al., 2022); [29] (Ayakdaş & Ağagündüz, 2023); and [30] (Devarshi et al., 2024).

▪ The Reference list has been entirely checked and renumbered accordingly.

Kind regards,

Ion Romulus SCOREI, Professor, PhD
